# BiSSL: Enhancing the Alignment Between Self-Supervised Pretraining and Downstream Fine-Tuning via Bilevel Optimization

**Gustav Wagner Zakarias**  *gwz@es.aau.dk*
*Aalborg University*
*Pioneer Centre for AI*

**Lars Kai Hansen**  *lkai@dtu.dk*
*Technical University of Denmark*
*Pioneer Centre for AI*

**Zheng-Hua Tan**  *zt@es.aau.dk*
*Aalborg University*
*Pioneer Centre for AI*

**Reviewed on OpenReview:** *https://openreview.net/forum?id=GQAGlqOpyA*

## Abstract

Models initialized from self-supervised pretraining may suffer from poor alignment with downstream tasks, limiting the extent to which subsequent fine-tuning can adapt relevant representations acquired during the pretraining phase. To mitigate this, we introduce BiSSL, a novel bilevel training framework that enhances the alignment of self-supervised pretrained models with downstream tasks by explicitly incorporating both the pretext and downstream tasks into a preparatory training stage prior to fine-tuning. BiSSL solves a bilevel optimization problem in which the lower-level adheres to the self-supervised pretext task, while the upper-level encourages the lower-level backbone to align with the downstream objective. The bilevel structure facilitates enhanced information sharing between the tasks, ultimately yielding a backbone model that is more aligned with the downstream task, providing a better initialization for subsequent fine-tuning. We propose a general training algorithm for BiSSL that is compatible with a broad range of pretext and downstream tasks. We demonstrate that our proposed framework significantly improves accuracy on the vast majority of a broad selection of image-domain downstream tasks, and that these gains are consistently retained across a wide range of experimental settings. In addition, exploratory alignment analyses further underpin that BiSSL enhances downstream alignment of pretrained representations.

## 1 Introduction

Self-supervised learning (SSL) leverages large amounts of heterogeneous unlabeled data to solve pretext tasks that encourage learning of general-purpose representations, which can be subsequently repurposed to solve downstream tasks. This has lead to strong performance across domains such as vision (Chen et al., 2020; Bardes et al., 2022; He et al., 2020; Grill et al., 2020; Caron et al., 2020; 2021; He et al., 2022; Oquab et al., 2024), language (Devlin et al., 2019; Lewis et al., 2019; Brown et al., 2020; He et al., 2021; Touvron et al., 2023), and audio (Schneider et al., 2019; Baevski et al., 2020; Hsu et al., 2021; Niizumi et al., 2021; Chung & Glass, 2018; Chung et al., 2019; Yadav et al., 2024). However, downstream tasks often involve narrower domains that demand more specialized features, which can result in the self-supervised pretrained representations being misaligned with these tasks. Pretrained features that benefit the downstream task may therefore initially appear irrelevant and thus risk being degraded or overwritten during fine-tuning, thereby

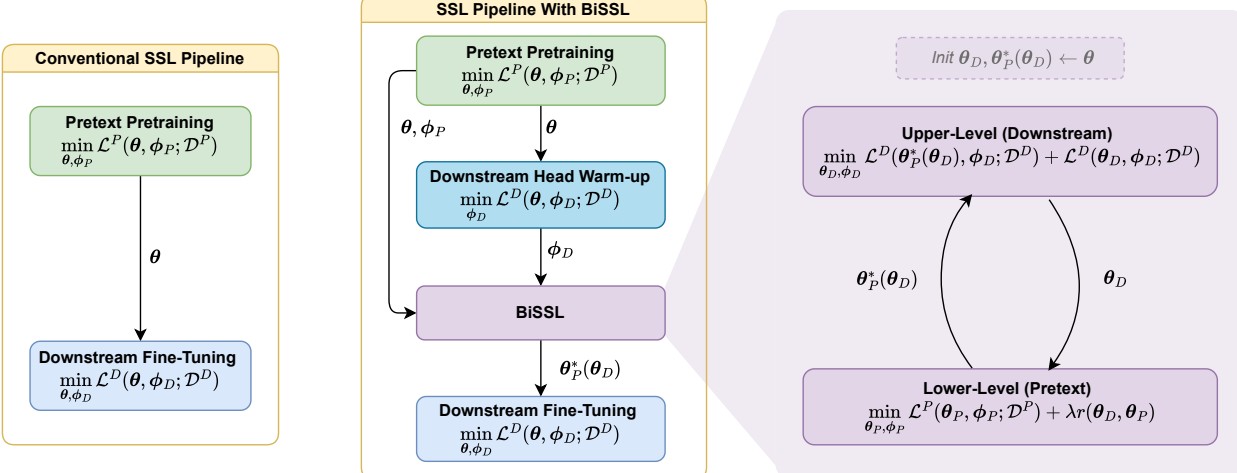

Figure 1: BiSSL introduces an intermediate training stage conducted after pretext pretraining. The symbols $\boldsymbol{\theta}$ and $\boldsymbol{\phi}$ represent backbone and task-specific attached head parameters, respectively. When they are transmitted to the respective subsequent training stages, they are used as initializations. The objectives $\mathcal{L}^P$, $\mathcal{L}^D$ represent the respective pretext pretraining and downstream fine-tuning objectives and $\mathcal{D}^P$, $\mathcal{D}^D$ the respective unlabeled pretext and labeled downstream datasets. We refer to Section 3 for further details.

constraining the achievable performance (Kumar et al., 2022; Zaiem et al., 2024; Liao et al., 2025; Wang et al., 2024; Boschini et al., 2022; Hinton & Salakhutdinov, 2006). Enhancing the alignment between the pretext and downstream tasks could address this issue, allowing the pretrained representations to be more optimally utilized during fine-tuning.

We suggest utilizing bilevel optimization (BLO) as a novel and effective approach for enhancing this alignment. BLO entails a main optimization problem constrained by the solution to a secondary problem that depends on the main problem's parameters. This hierarchical setup captures and leverages interdependencies between problems and has shown strong potential in deep learning settings that involve joint optimization of multiple coupled objectives (Zhang et al., 2023a) such as parameter pruning (Zhang et al., 2022b), invariant risk minimization (Arjovsky et al., 2019; Zhang et al., 2023b), meta-learning (Rajeswaran et al., 2019; Finn et al., 2017), adversarial robustness (Zhang et al., 2021), hyper-parameter optimization (Franceschi et al., 2018), and coreset selection (Borsos et al., 2020).

In this study, we propose BiSSL, a novel bilevel training framework that improves the alignment of self-supervised pretrained backbone models with downstream tasks. BiSSL introduces a preparatory training stage prior to fine-tuning that solves a bilevel optimization problem in which the pretext and downstream task objectives serve as the lower- and upper-level objectives, respectively. This bilevel setup facilitates enhanced information sharing, enabling the upper-level to guide the pretext task optimization in refining its representations to better align with the downstream objective, ultimately providing a more effective backbone initialization for fine-tuning. Figure 1 depicts how BiSSL integrates into the standard SSL pipeline. We introduce a training algorithm that alternates between optimizing the two objectives within BiSSL, which is agnostic to the choice of pretext and downstream task. Our method makes no additional assumptions about how models are pretrained, allowing compatibility with standard off-the-shelf pretrained backbones. Through experiments using both the SimCLR (Chen et al., 2020) and Bootstrap Your Own Latent (Grill et al., 2020) pretext tasks to pretrain ResNet-50 backbones (He et al., 2016) on the ImageNet-1K dataset (Deng et al., 2009), we demonstrate that BiSSL significantly improves downstream performance across most of 12 downstream image classification datasets, while also significantly improving accuracy within object detection and semantic segmentation. A suite of extended evaluations further shows that the gains remain consistent, and exploratory alignment studies of backbones highlight that BiSSL improves downstream alignment. Code and pretrained model weights are publicly available at `https://github.com/GustavWZ/bissl/`.

## 2 Related Work

**Bilevel Optimization in Self-Supervised Learning**   Bilevel optimization (BLO) refers to a constrained optimization problem, where the constraint itself is a solution to another optimization problem that depends on the parameters of the "main" optimization problem. The general BLO problem is formulated as

$$\min_{\boldsymbol{\xi}} f(\boldsymbol{\xi}, \boldsymbol{\psi}^*(\boldsymbol{\xi})) \quad \text{s.t.} \quad \boldsymbol{\psi}^*(\boldsymbol{\xi}) \in \operatorname*{argmin}_{\boldsymbol{\psi}} g(\boldsymbol{\xi}, \boldsymbol{\psi}), \tag{1}$$

where $f$ and $g$ are referred to as the upper-level and lower-level objectives, respectively. While the lower objective $g$ has knowledge of the parameters $\boldsymbol{\xi}$ from the upper-level objective, the upper-level objective $f$ possesses full information of the lower objective $g$ itself through its dependence on the lower-level solution $\boldsymbol{\psi}^*(\boldsymbol{\xi})$. Some works have incorporated bilevel optimization within self-supervised learning. Gupta et al. (2022) suggest formulating the contrastive self-supervised pretext task as a bilevel optimization problem, dedicating the upper-level and lower-level objectives for updating the backbone and projection head parameters respectively. Other frameworks such as the Local and Global (LoGo) (Zhang et al., 2022a) and Only Self-Supervised Learning (OSSL) (Boonlia et al., 2022) utilize auxiliary models, wherein the lower-level objective optimizes the parameters of the auxiliary model, while the upper-level objective is dedicated to training the feature extraction model. MetaMask (Li et al., 2022b) introduces a meta-learning based approach, where the upper-level learns masks that filter out irrelevant information from inputs that are provided to a lower-level self-supervised contrastive pretext task. BLO-SAM (Zhang et al., 2024) is tailored towards fine-tuning the segment anything model (SAM) (Kirillov et al., 2023) by interchangeably alternating between learning (upper-level) prompt embeddings and fine-tuning the (lower-level) segmentation model. The aforementioned frameworks integrate bilevel optimization into *either* the pretraining or fine-tuning stage exclusively and are tailored towards specific pretext or downstream tasks. In contrast, our proposed BiSSL employs a BLO problem that comprehensively incorporates *both* training stages of pretext pretraining and downstream fine-tuning, while not being confined to any specific type of pretext or downstream task.

**Priming Pretrained Backbones Prior To Fine-Tuning**   Previous works have also demonstrated that downstream performance can be enhanced by introducing techniques that modify the backbone between the pretraining and fine-tuning stages. Contrastive Initialization (COIN) (Pan et al., 2022) introduces a supervised contrastive loss to be utilized on backbones pretrained with contrastive SSL techniques. Noisy-Tune (Wu et al., 2022) perturbs the pretrained backbone with tailored noise before fine-tuning. Speaker-invariant clustering (Spin) (Chang et al., 2023) utilizes speaker disentanglement and vector quantization for improving speech representations for speech signal specific downstream tasks. RIFLE (Li et al., 2020) conducts multiple fine-tuning sessions sequentially, where the attached downstream specific layers are re-initialized in between every session. Unlike BiSSL, these techniques either do not incorporate knowledge of both the pretext task and downstream task objectives and their relationship, or do so only implicitly.

## 3 Proposed Method

### 3.1 Notation

We denote the unlabeled pretext dataset $\mathcal{D}^P = \{\mathbf{z}_k\}_{k=1}^{|\mathcal{D}^P|}$ and labeled downstream dataset $\mathcal{D}^D = \{\mathbf{x}_l, \mathbf{y}_l\}_{l=1}^{|\mathcal{D}^D|}$, respectively, where $\mathbf{z}_k, \mathbf{x}_l \in \mathbb{R}^N$. Let $f_{\boldsymbol{\theta}} : \mathbb{R}^N \to \mathbb{R}^M$ denote a feature extracting backbone with trainable parameters $\boldsymbol{\theta}$ and $h_{\boldsymbol{\phi}}^T : \mathbb{R}^M \to \mathbb{R}^{Q_T}$ a task specific head with trainable parameters $\boldsymbol{\phi}$. Given pretext and downstream models $h_{\boldsymbol{\phi}_P}^P \circ f_{\boldsymbol{\theta}_P}$ and $h_{\boldsymbol{\phi}_D}^D \circ f_{\boldsymbol{\theta}_D}$ with $\boldsymbol{\theta}_P, \boldsymbol{\theta}_D \in \mathbb{R}^L$, we denote the pretext and downstream training objectives $\mathcal{L}^P(\boldsymbol{\theta}_P, \boldsymbol{\phi}_P; \mathcal{D}^P)$ and $\mathcal{L}^D(\boldsymbol{\theta}_D, \boldsymbol{\phi}_D; \mathcal{D}^D)$, respectively. To simplify notation, we omit the dataset specification from the training objectives, e.g. $\mathcal{L}^D(\boldsymbol{\theta}_D, \boldsymbol{\phi}_D) := \mathcal{L}^D(\boldsymbol{\theta}_D, \boldsymbol{\phi}_D; \mathcal{D}^D)$.

### 3.2 Optimization Problem Formulation

In standard SSL, a single backbone with parameters $\boldsymbol{\theta}$ is first trained to minimize the pretext objective $\mathcal{L}^P(\boldsymbol{\theta}, \boldsymbol{\phi}_P)$, yielding a parameter configuration $\boldsymbol{\theta}^*$ that subsequently is used as an initialization for fine-tuning on the downstream objective $\mathcal{L}^D(\boldsymbol{\theta}, \boldsymbol{\phi}_D)$. To better align the self-supervised pretrained initialization

with the downstream task, we employ BiSSL as a preparatory training stage to fine-tuning that incorporates the pretext and downstream task objectives into a joint bilevel optimization problem. BiSSL uses two distinct sets of backbone parameters $\boldsymbol{\theta}_P$ and $\boldsymbol{\theta}_D$, corresponding to the pretext and downstream tasks, that are treated as separate but strongly correlated parameter vectors. The bilevel optimization problem is formulated as

$$\min_{\boldsymbol{\theta}_D, \boldsymbol{\phi}_D} \mathcal{L}^D\left(\boldsymbol{\theta}_P^*\left(\boldsymbol{\theta}_D\right), \boldsymbol{\phi}_D\right) + \mathcal{L}^D\left(\boldsymbol{\theta}_D, \boldsymbol{\phi}_D\right) \tag{2}$$

$$\text{s.t. } \boldsymbol{\theta}_P^*\left(\boldsymbol{\theta}_D\right) \in \operatorname*{argmin}_{\boldsymbol{\theta}_P} \min_{\boldsymbol{\phi}_P} \mathcal{L}^P\left(\boldsymbol{\theta}_P, \boldsymbol{\phi}_P\right) + \lambda r(\boldsymbol{\theta}_P, \boldsymbol{\theta}_D), \tag{3}$$

with $r$ being some convex regularization objective weighted by $\lambda \in \mathbb{R}_+$ enforcing similarity between $\boldsymbol{\theta}_P$ and $\boldsymbol{\theta}_D$. The upper-level training objective in (2) is tasked with minimizing the downstream task objective $\mathcal{L}^D$ and the lower-level objective in (3) aims to minimize the pretext task objective $\mathcal{L}^P$ while also ensuring its backbone remains similar to the upper-level backbone. As seen in the left term of (2), the backbone parameters $\boldsymbol{\theta}_P^*(\boldsymbol{\theta}_D)$ are substituted into the downstream training objective, resembling how backbones conventionally are transferred after self-supervised pretraining. The lower-level's dependence on the regularization objective $r$ enables the upper-level to observe how changes in its own backbone parameters $\boldsymbol{\theta}_D$ influence the lower-level solution $\boldsymbol{\theta}_P^*(\boldsymbol{\theta}_D)$, thereby allowing the upper-level to indirectly guide the pretext task optimization toward a lower-level backbone that better complies with the downstream task. The second term of (2) aids convergence of the upper-level objective during training by acting as a regularizer that constrains $\boldsymbol{\theta}_D$ to itself also align with the downstream task. Without this term, $\boldsymbol{\theta}_D$ would function only as an indirect controller of the lower-level, potentially leading to degenerate solutions.

### 3.3 Upper-level Derivative

Given the upper-level objective $F(\boldsymbol{\theta}_D, \boldsymbol{\phi}_D) := \mathcal{L}^D(\boldsymbol{\theta}_P^*(\boldsymbol{\theta}_D), \boldsymbol{\phi}_D) + \mathcal{L}^D(\boldsymbol{\theta}_D, \boldsymbol{\phi}_D)$ from (2), its derivative with respect to $\boldsymbol{\theta}_D$ is given by

$$\frac{\mathrm{d}F}{\mathrm{d}\boldsymbol{\theta}_D} = \underbrace{\frac{\mathrm{d}\boldsymbol{\theta}_P^*(\boldsymbol{\theta}_D)}{\mathrm{d}\boldsymbol{\theta}_D}^T}_{\text{IJ}} \nabla_{\boldsymbol{\theta}} \mathcal{L}^D(\boldsymbol{\theta}, \boldsymbol{\phi}_D)|_{\boldsymbol{\theta}=\boldsymbol{\theta}_P^*(\boldsymbol{\theta}_D)} + \nabla_{\boldsymbol{\theta}} \mathcal{L}^D(\boldsymbol{\theta}, \boldsymbol{\phi}_D)|_{\boldsymbol{\theta}=\boldsymbol{\theta}_D}. \tag{4}$$

Due to the dependence of the lower-level solution on the upper-level parameters, the first term of (4) includes the Jacobian of the implicit function $\boldsymbol{\theta}_P^*(\boldsymbol{\theta}_D)$, referred to as the implicit Jacobian (IJ). To simplify notation, we let $\nabla_{\boldsymbol{\xi}} h(\boldsymbol{\xi})|_{\boldsymbol{\xi}=\boldsymbol{\psi}} := \nabla_{\boldsymbol{\xi}} h(\boldsymbol{\psi})$ when it is clear from context which variables are differentiated with respect to. Following an approach similar to Rajeswaran et al. (2019), with details on the derivations and underlying assumptions outlined in Section A.1, the IJ in (4) can be explicitly expressed as

$$\frac{\mathrm{d}\boldsymbol{\theta}_P^*(\boldsymbol{\theta}_D)}{\mathrm{d}\boldsymbol{\theta}_D}^T = -\nabla^2_{\boldsymbol{\theta}_D \boldsymbol{\theta}_P} r(\boldsymbol{\theta}_P^*(\boldsymbol{\theta}_D), \boldsymbol{\theta}_D) \left[\nabla^2_{\boldsymbol{\theta}_P}\left(\frac{1}{\lambda} \mathcal{L}^P(\boldsymbol{\theta}_P^*(\boldsymbol{\theta}_D), \boldsymbol{\phi}_P) + r(\boldsymbol{\theta}_P^*(\boldsymbol{\theta}_D), \boldsymbol{\theta}_D)\right)\right]^{-1}. \tag{5}$$

A common convex regularization objective, which will also be the choice in the subsequent experiments of this work, is $r(\boldsymbol{\xi}, \boldsymbol{\psi}) = \frac{1}{2}\|\boldsymbol{\xi} - \boldsymbol{\psi}\|_2^2$. Using this regularization objective simplifies (5) down to

$$\frac{\mathrm{d}\boldsymbol{\theta}_P^*(\boldsymbol{\theta}_D)}{\mathrm{d}\boldsymbol{\theta}_D}^T = \left[\frac{1}{\lambda} \nabla^2_{\boldsymbol{\theta}_P} \mathcal{L}^P(\boldsymbol{\theta}_P^*(\boldsymbol{\theta}_D), \boldsymbol{\phi}_P) + I_L\right]^{-1}, \tag{6}$$

where $I_L$ is the $L \times L$-dimensional identity matrix. Hence the upper-level derivative (4) can be expressed as

$$\frac{\mathrm{d}F}{\mathrm{d}\boldsymbol{\theta}_D} = \left[\frac{1}{\lambda} \nabla^2_{\boldsymbol{\theta}_P} \mathcal{L}^P(\boldsymbol{\theta}_P^*(\boldsymbol{\theta}_D), \boldsymbol{\phi}_P) + I_L\right]^{-1} \nabla_{\boldsymbol{\theta}_P} \mathcal{L}^D(\boldsymbol{\theta}_P^*(\boldsymbol{\theta}_D), \boldsymbol{\phi}_D) + \nabla_{\boldsymbol{\theta}_D} \mathcal{L}^D(\boldsymbol{\theta}_D, \boldsymbol{\phi}_D). \tag{7}$$

As shown in (7), the pretext objective is explicitly incorporated into the upper-level gradient, causing the upper-level updates to be influenced by how its changes affect the lower-level.

The inverse Hessian-vector product in (7) is computationally infeasible to calculate directly, so it is approximated using the conjugate gradient (CG) method (Nazareth, 2009; Shewchuk, 1994). We employ a layer-wise implementation of the CG method based on that of Rajeswaran et al. (2019) and refer to their work for more details on applying CG in a deep learning setup with BLO. While CG has proven to be an effective approach for approximating the inverse Hessian-vector products in previous works (Pedregosa, 2016; Zhang et al., 2021; Rajeswaran et al., 2019), it still introduces some computation and memory overhead due to its need for evaluations of multiple Hessian vector products. Future work could explore alternative established methods for upper-level gradient approximation (Zhang et al., 2023a; Choe et al., 2023; Liu et al., 2024) that offer greater computational efficiency. We report the computation times of BiSSL alongside along pretraining and fine-tuning and discuss the Hessian vector product calculation overhead in Section C.1.

With an explicit expression of the IJ in (6), we can interpret the impact of the scaling factor $\lambda$ from (3) and (7): When $\lambda$ is very large, the dependence of the lower-level objective on the upper-level parameters $\boldsymbol{\theta}_D$ is correspondingly very large. This effectively drives the lower-level backbone parameters toward the trivial solution $\boldsymbol{\theta}_P^*(\boldsymbol{\theta}_D) = \boldsymbol{\theta}_D$. Meanwhile, the IJ in (6) approximately equals $I_L$, thereby diminishing the influence of the lower-level objective on the upper-level gradient in (7). This roughly makes the task of the upper-level equivalent to conventional fine-tuning. Conversely, if $\lambda$ is very small, the lower-level objective in (3) effectively defaults to conventional pretext task training. Additionally, the implicit Jacobian in (6) would consist of numerically tiny entries, making the optimization of the first term in the upper-level objective in (2) equivalent to linear probing of the downstream head on the frozen pretext backbone $\boldsymbol{\theta}_P^*(\boldsymbol{\theta}_D)$. Thus, an intermediate value of $\lambda$ is expected to yield an optimization that balances the contribution from both levels.

### 3.4 Training Algorithm and Pipeline

---

**Algorithm 1** BiSSL Training Algorithm

---

1: **Input:** Backbone and head initializations $\boldsymbol{\theta}$, $\boldsymbol{\phi}_P$, $\boldsymbol{\phi}_D$. Training objectives $\mathcal{L}^P$, $\mathcal{L}^D$. Optimizers $\text{opt}_P$, $\text{opt}_D$. Regularization Weight $\lambda \in \mathbb{R}_+$. Number of training stage alternations $T \in \mathbb{N}$ with upper and lower-level iterations $N_U, N_L \in \mathbb{N}$.

2: Initialize $\boldsymbol{\theta}_P \leftarrow \boldsymbol{\theta}$ and $\boldsymbol{\theta}_D \leftarrow \boldsymbol{\theta}$.

3: **for** $t = 1, \ldots, T$ **do**
4:      **for** $n = 1, \ldots, N_L$ **do**                                           ▷ Lower-level
5:          Compute $\mathbf{g}_{\boldsymbol{\phi}_P} = \nabla_{\boldsymbol{\phi}} \mathcal{L}^P(\boldsymbol{\theta}_P, \boldsymbol{\phi})|_{\boldsymbol{\phi}=\boldsymbol{\phi}_P}$ .
6:          Compute $\mathbf{g}_{\boldsymbol{\theta}_P} = \nabla_{\boldsymbol{\theta}} \mathcal{L}^P(\boldsymbol{\theta}, \boldsymbol{\phi}_P)|_{\boldsymbol{\theta}=\boldsymbol{\theta}_P} + \lambda(\boldsymbol{\theta}_P - \boldsymbol{\theta}_D)$.
7:          Update $\boldsymbol{\phi}_P \leftarrow \text{opt}_P(\boldsymbol{\phi}_P, \mathbf{g}_{\boldsymbol{\phi}_P})$ and $\boldsymbol{\theta}_P \leftarrow \text{opt}_P(\boldsymbol{\theta}_P, \mathbf{g}_{\boldsymbol{\theta}_P})$.

8:      **for** $n = 1, \ldots, N_U$ **do**                                           ▷ Upper-level
9:          Compute $\mathbf{g}_{\boldsymbol{\phi}_D} = \nabla_{\boldsymbol{\phi}} \mathcal{L}^D(\boldsymbol{\theta}_P, \boldsymbol{\phi})|_{\boldsymbol{\phi}=\boldsymbol{\phi}_D} + \nabla_{\boldsymbol{\phi}} \mathcal{L}^D(\boldsymbol{\theta}_D, \boldsymbol{\phi})|_{\boldsymbol{\phi}=\boldsymbol{\phi}_D}$.
10:          Compute $\mathbf{v} = \nabla_{\boldsymbol{\theta}} \mathcal{L}^D(\boldsymbol{\theta}, \boldsymbol{\phi}_D)|_{\boldsymbol{\theta}=\boldsymbol{\theta}_P}$.
11:          Approximate $\mathbf{v}_{\text{IJ}} \approx \left[\frac{1}{\lambda}\nabla_{\boldsymbol{\theta}}^2 \mathcal{L}^P(\boldsymbol{\theta}, \boldsymbol{\phi}_P)|_{\boldsymbol{\theta}=\boldsymbol{\theta}_P} + I_L\right]^{-1}\mathbf{v}$.          ▷ Use CG to approximate
12:          Compute $\mathbf{g}_{\boldsymbol{\theta}_D} = \mathbf{v}_{\text{IJ}} + \nabla_{\boldsymbol{\theta}} \mathcal{L}^D(\boldsymbol{\theta}, \boldsymbol{\phi}_D)|_{\boldsymbol{\theta}=\boldsymbol{\theta}_D}$.
13:          Update $\boldsymbol{\phi}_D \leftarrow \text{opt}_D(\boldsymbol{\phi}_D, \mathbf{g}_{\boldsymbol{\phi}_D})$ and $\boldsymbol{\theta}_D \leftarrow \text{opt}_D(\boldsymbol{\theta}_D, \mathbf{g}_{\boldsymbol{\theta}_D})$.

14: **Return:** Backbone Parameters $\boldsymbol{\theta}_P$.

---

Our proposed training algorithm iteratively alternates between solving the lower-level (3) and upper-level (2) optimization problems in BiSSL. The lower-level training optimizes the pretext task objective, while additionally including the gradient of the regularization term $r$, complying with (3). The upper-level stage computes gradients with respect to the backbone, corresponding to the left-hand term in (7), which are approximated using the CG method. Algorithm 1 details the proposed training procedure. We remark that allowing $N_U > 1$ deviates from conventional BLO training setups. However, as we document in Sec-

tion C.3, this modification proved more efficient and beneficial for downstream performance. The algorithm is applicable to any common pretext and downstream tasks.

Figure 1 illustrates the proposed training pipeline with BiSSL alongside the conventional SSL pipeline. Pretext pretraining on the unlabeled dataset $\mathcal{D}^P$ provides initializations of $\boldsymbol{\theta}$ and $\boldsymbol{\phi}_P$, after which the downstream head $\boldsymbol{\phi}_D$ is fitted on top of the frozen backbone $\boldsymbol{\theta}$ using the downstream dataset $\mathcal{D}^D$. BiSSL training is then conducted as outlined in Algorithm 1, yielding an updated backbone $\boldsymbol{\theta}_P^*(\boldsymbol{\theta}_D)$, which serves as the initialization for final supervised fine-tuning on the downstream task. We emphasize that the objective $\mathcal{L}^D$ in the upper-level of BiSSL is identical to the downstream fine-tuning objective, and similarly that the objective $\mathcal{L}^P$ in the lower-level is identical to the pretext pretraining objective.

Section A.1 notes that $\boldsymbol{\theta}_P^*(\boldsymbol{\theta}_D)$ must satisfy the stationary condition $\nabla_{\boldsymbol{\theta}}\big(\mathcal{L}^P(\boldsymbol{\theta}, \boldsymbol{\phi}_P) + \lambda r(\boldsymbol{\theta}, \boldsymbol{\theta}_D)\big)|_{\boldsymbol{\theta}=\boldsymbol{\theta}_P^*(\boldsymbol{\theta}_D)} = \mathbf{0}$ to justify the explicit expression of the IJ in (6). This requirement naturally aligns with using a backbone that has already converged from self-supervised pretraining, as such an initialization roughly satisfies the stationary condition at the outset and also makes the smoothness assumption on $\mathcal{L}^P$, mentioned in Section A.1, more reasonable. A similar consideration applies to $\boldsymbol{\phi}_D$, since a randomly initialized downstream head can lead to rapid early updates to $\boldsymbol{\theta}_D$ during fine-tuning, potentially violating the stationary assumption due to its coupling with $\boldsymbol{\theta}_P$ through the regularization objective $r$.

## 4 Experiments and Results

### 4.1 Datasets

The ImageNet-1K (Deng et al., 2009) dataset, devoid of labels, is used for self-supervised pretraining throughout the main experiments. For downstream fine-tuning and evaluation, we leverage a varied set of natural image classification datasets that encompass a wide array of tasks, including general image classification, fine-grained recognition across species and objects, scene understanding, and texture categorization. The datasets include Food 101 (Bossard et al., 2014), CIFAR10 (Krizhevsky, 2012), CIFAR100 (Krizhevsky, 2012), Caltech-UCSD Birds-200-2011 (CUB200) (Wah et al., 2011), SUN397 scene dataset (Xiao et al., 2010), StanfordCars (Yang et al., 2015), FGVC Aircraft (Maji et al., 2013), PASCAL VOC 2007 (Everingham et al., 2010), Describable Textures Dataset (DTD) (Cimpoi et al., 2014), Oxford-IIIT Pets (Parkhi et al., 2012), Caltech-101 (Li et al., 2022a) and Oxford 102 Flowers (Nilsback & Zisserman, 2008). Additionally, we use the combined VOC07+12 (Everingham et al., 2010) dataset for both object detection and semantic segmentation, and the fine-grained partition of Cityscapes (Cordts et al., 2016) for semantic segmentation. All downstream datasets are split into training, validation, and test partitions, with details on how these assignments are made in Section B.1.

### 4.2 Implementation Details

#### 4.2.1 Baseline Setup

We first outline each stage of the baseline setup following the standard SSL pipeline on the left of Figure 1.

**Pretext Task Training** We conduct experiments using two different types of pretext tasks: SimCLR (Chen et al., 2020) and Bootstrap Your Own Latent (BYOL) (Grill et al., 2020). For SimCLR we use a temperature of 0.5 and a ResNet-50 (He et al., 2016) backbone. On top of the backbone, a projection head is applied, consisting of two fully connected layers with batch normalization (Ioffe & Szegedy, 2015) and ReLU (Agarap, 2018) followed by a single linear layer. Each layer consists of 256 neurons. For BYOL, we use constant target decay rate of 0.9995, with the backbone and projection head architectures identical to those of SimCLR. The additional BYOL-specific prediction head uses an architecture identical to the projection head. The remaining details in this paragraph apply to both SimCLR and BYOL.

The image augmentation scheme follows the approach used in Bardes et al. (2022), with minor modifications: The image size is set to $96 \times 96$ instead of $224 \times 224$, and the minimal ratio of the random crop is adjusted accordingly to 0.5 instead of 0.08. The implementation of the LARS optimizer (You et al., 2017) from Bardes et al. (2022) is employed, with a "trust" coefficient of 0.001, a weight decay of $10^{-6}$ and a momentum of

0.9. The learning rate increases linearly during the first 10 epochs, reaching a peak base learning rate of 4.8, followed by a cosine decay with no restarts (Loshchilov & Hutter, 2017) for the remaining epochs. A batch size of 1024 is used and, unless otherwise specified, pretraining is conducted for 500 epochs.

**Fine-Tuning on the Downstream Task**  For downstream fine-tuning, a single linear layer is attached to the output of the pretrained backbone. The training procedure utilizes the cross-entropy loss, the SGD optimizer with a momentum of 0.9, and a cosine decaying learning rate scheduler without restarts (Loshchilov & Hutter, 2017). Fine-tuning is conducted for 400 epochs with a batch size of 256. An augmentation scheme similar to the fine-tuning augmentation scheme in Bardes et al. (2022) is employed during training, where images are center cropped and resized to $96 \times 96$ pixels with a minimal crop ratio of 0.5, followed by random horizontal flips. Learning rates and weight decays for each downstream datasets are selected via a random hyper-parameter grid search. After hyperparameters are determined, we train 10 models with different random seeds, and report the mean and standard deviation of the top-1 test accuracies. Details on the hyperparameter search and evaluation protocol are provided in Section B.2.

### 4.2.2 BiSSL Setup

This section detail each stage of the training pipeline involving BiSSL, as illustrated on the right of Figure 1.

**Pretext Task Training**  The backbone $\boldsymbol{\theta}$ and projection head $\boldsymbol{\phi}_P$ are initialized by self-supervised pre-training using a setup identical to the baseline pretext task training setup detailed in Section 4.2.1. This enables reusing the pretrained backbones from the baseline experiments.

**Downstream Head Warm-up**  The training setup for the downstream head warm-up closely mirrors the fine-tuning setup of Section 4.2.1. The main difference is that only the linear downstream head is fitted on top of the now frozen backbone obtained from the pretext warm-up. Learning rates and weight decays are initially selected based on those listed in Table 5, with adjustments made as needed when preliminary testing indicated a potential for improved convergence. These values are provided in Table 6 of Appendix B. The authors recognize that more optimal hyper-parameter values may exist and leave further exploration of this for future refinement. The downstream head warm-up is run for 20 epochs with a constant learning rate.

**Lower-level of BiSSL**  The training configuration for the lower-level primarily follows the setup described for pretext pretraining in Section 4.2.1, with the modifications outlined here. As specified in (3), the lower-level loss function is the sum of the pretext task objective $\mathcal{L}^P$ (e.g. the NT-Xent loss for SimCLR (Chen et al., 2020)) and the regularization term $r(\boldsymbol{\theta}_P, \boldsymbol{\theta}_D) = \frac{1}{2}||\boldsymbol{\theta}_P - \boldsymbol{\theta}_D||_2^2$, aligning with Algorithm 1. Based on early experiments, the regularization weight $\lambda = 0.001$ was selected, as it appeared to strike a well-balanced compromise between the convergence rates of both the lower- and upper-level objectives. The lower-level is trained for the equivalent of approximately 8 conventional pretraining epochs, with further details provided in the composite configuration details paragraph. Accordingly the linear learning rate scheduler warm-up is adjusted to range over $10 \cdot N_L$ training steps, where $N_L$ is the number of lower-level iterations in Algorithm 1.

**Upper-level of BiSSL**  The upper-level training stage largely mirrors the downstream training setup described in Section 4.2.1, and we again only list the differences. The weight decays and base learning rates are set to match those obtained from the downstream head warm-up detailed in Table 6 of Appendix B. To approximate the upper-level gradient in (7), the conjugate gradient method (Nazareth, 2009; Shewchuk, 1994) is employed, with implementation details covered in Section B.3.

**Composite Configuration Details of BiSSL**  As outlined in Algorithm 1, both lower- and upper-level backbone parameters $\boldsymbol{\theta}_P$ and $\boldsymbol{\theta}_D$ are initialized with the backbone parameters obtained during the pretext warm-up, and the training procedure alternates between solving the lower- and upper-level optimization problems. In this experimental setup, the lower-level performs $N_L = 20$ gradient steps before alternating to the upper-level, which then conducts $N_U = 8$ gradient steps. A total of $T = 500$ training stage alternations are executed. With the ImageNet dataset and the current batch size of 1024, there are a total of 1251 training batches without replacement. Consequently, the $T = 500$ training stage alternations correspond

Table 1: Comparison of classification accuracies obtained through fine-tuning without and with preparatory BiSSL training. Statistically significant improvements over the counterpart are highlighted in bold.

| | Food | CIFAR10 | CIFAR100 | CUB200 | SUN397 | Cars | Aircrafts | VOC07 | DTD | Pets | Caltech101 | Flowers |
|---|---|---|---|---|---|---|---|---|---|---|---|---|
| **SimCLR:** | | | | | | | | | | | | |
| FT | $75.0 \pm 0.1$ | $96.1 \pm 0.2$ | $79.5 \pm 0.2$ | $49.6 \pm 0.3$ | $49.3 \pm 0.2$ | $77.9 \pm 0.3$ | $52.0 \pm 1.0$ | $71.0 \pm 0.1$ | $60.4 \pm 0.9$ | $73.5 \pm 0.7$ | $85.8 \pm 0.6$ | $82.6 \pm 0.3$ |
| BiSSL+FT | $\mathbf{76.2 \pm 0.1}$ | $\mathbf{96.3 \pm 0.1}$ | $\mathbf{79.8 \pm 0.2}$ | $\mathbf{55.7 \pm 0.2}$ | $\mathbf{50.9 \pm 0.1}$ | $77.9 \pm 0.3$ | $\mathbf{55.9 \pm 0.3}$ | $\mathbf{71.5 \pm 0.1}$ | $\mathbf{64.2 \pm 0.4}$ | $\mathbf{78.3 \pm 0.3}$ | $\mathbf{87.2 \pm 0.3}$ | $\mathbf{84.1 \pm 0.2}$ |
| *Avg Diff* | **+1.2** | **+0.2** | **+0.4** | **+6.1** | **+1.6** | 0.0 | **+3.9** | **+0.5** | **+3.8** | **+4.8** | **+1.4** | **+1.5** |
| **BYOL:** | | | | | | | | | | | | |
| FT | $75.3 \pm 0.3$ | $96.4 \pm 0.1$ | $80.1 \pm 0.2$ | $52.7 \pm 0.4$ | $47.9 \pm 0.3$ | $76.9 \pm 0.2$ | $51.6 \pm 0.7$ | $69.3 \pm 0.1$ | $59.5 \pm 0.4$ | $77.9 \pm 0.3$ | $86.6 \pm 0.4$ | $82.1 \pm 0.5$ |
| BiSSL+FT | $\mathbf{76.2 \pm 0.1}$ | $96.4 \pm 0.1$ | $\mathbf{80.4 \pm 0.1}$ | $\mathbf{57.1 \pm 0.3}$ | $49.6 \pm 0.1$ | $\mathbf{77.5 \pm 0.2}$ | $\mathbf{57.2 \pm 0.5}$ | $\mathbf{70.8 \pm 0.1}$ | $\mathbf{61.7 \pm 0.2}$ | $\mathbf{80.1 \pm 0.3}$ | $\mathbf{87.2 \pm 0.2}$ | $\mathbf{85.4 \pm 0.2}$ |
| *Avg Diff* | **+0.9** | 0.0 | **+0.3** | **+4.4** | **+1.7** | **+0.6** | **+5.6** | **+1.5** | **+2.2** | **+2.2** | **+0.6** | **+3.3** |

to roughly 8 conventional pretext epochs, a negligible additional training load compared to the 500 pretext epochs used for the full pretraining process. Section B.4 outlines further details on how data batches are handled during training. Lastly, gradient normalization is employed on gradients exceeding $\ell_2$-norms of 10.

**Fine-Tuning on the Downstream Task**   Subsequent downstream fine-tuning is conducted in a manner identical to that described in the "Fine-Tuning on the Downstream Task" paragraph of section 4.2.1. Table 7 in Appendix B lists the considered optimal hyper-parameter configurations for each dataset.

## 4.3   Downstream Task Performance

Table 1 reports means and standard deviations of top-1 classification accuracies (or 11-point mAP on the VOC07 dataset), comparing results obtained from direct fine-tuning the pretrained backbone with those achieved by applying BiSSL prior to fine-tuning. We mark statistically significantly different accuracies with bold-font, with details on how the statistical significance tests are executed in Section B.6. Table 11 of Section C.2 outlines the corresponding top-5 accuracies. For both pretext tasks, BiSSL significantly improves downstream performance on 11 out of 12 datasets, with no single result showing a decline in performance. Moreover, accuracies from BiSSL exhibit lower or similar variances than the baseline. Smaller gains on tasks like CIFAR10, VOC07, and Caltech101 may reflect their coarse-grained nature and input distributions being more similar to the pretext task, whereas fine-grained tasks with more distinct input distributions that require more specialized representations like CUB200, Aircrafts, and Pets, exhibit greater gains from BiSSL.

### 4.3.1   Object Detection and Semantic Segmentation

We further assess performance in the context of object detection and semantic segmentation. Similar to the setup of Bardes et al. (2022); He et al. (2020), we use for object detection a Faster-RCNN style architecture with a ResNet-50 C4 backbone (Ren et al., 2015) on the VOC07+12 (Everingham et al., 2010) dataset and report the standard $AP_{50}$ metric. For semantic segmentation tasks, we follow the setup of He et al. (2020); Grill et al. (2020) and use a FCN-16 (Long et al., 2015b) based structure on top of the pretrained ResNet-50 backbone and evaluate on the VOC07+12 (Everingham et al., 2010) and CityScapes (Cordts et al., 2016) datasets, reporting the mean intersection over union (mIoU) scores. We refer to Section B.7 for complete implementation details. The results in Table 2 support the previous conclusions, indicating that BiSSL overall significantly improves downstream performance.

Table 2: Comparison of $AP_{50}$ scores for object detection and mIoU scores for semantic segmentation obtained by fine-tuning without and with preparatory BiSSL training. Scores statistically significantly higher from their counterparts are marked in bold.

| | Detection ($AP_{50}$) VOC07+12 | Semantic Seg. (mIoU) VOC07+12 | CityScapes |
|---|---|---|---|
| **SimCLR:** | | | |
| FT | $53.1 \pm 0.3$ | $72.7 \pm 0.6$ | $47.8 \pm 0.4$ |
| BiSSL+FT | $\mathbf{54.4 \pm 0.4}$ | $\mathbf{74.4 \pm 0.6}$ | $\mathbf{50.7 \pm 0.4}$ |
| *Avg Diff* | **+1.3** | **+1.7** | **+2.9** |
| **BYOL:** | | | |
| FT | $52.6 \pm 0.7$ | $71.7 \pm 0.6$ | $48.6 \pm 0.5$ |
| BiSSL+FT | $\mathbf{53.5 \pm 0.3}$ | $\mathbf{73.2 \pm 0.4}$ | $\mathbf{49.7 \pm 0.4}$ |
| *Avg Diff* | **+0.9** | **+1.5** | **+1.1** |

Table 3: Comparison of downstream classification accuracies across baseline methods detailed in Section B.9. For each dataset, the highest average accuracy is marked in bold, and the second-highest is underlined.

| | Pets | DTD | VOC07 | Flowers | CUB200 |
|---|---|---|---|---|---|
| Random Initialization (B.9.1) | $53.9 \pm 0.8$ | $35.6 \pm 0.5$ | $43.7 \pm 0.3$ | $43.7 \pm 0.5$ | $25.7 \pm 0.4$ |
| Linear Probing (B.9.1) | $58.4 \pm 0.2$ | $56.3 \pm 0.4$ | $63.4 \pm 0.1$ | $64.5 \pm 0.1$ | $18.5 \pm 0.1$ |
| FT | $73.5 \pm 0.7$ | $60.4 \pm 0.9$ | $71.0 \pm 0.1$ | $82.6 \pm 0.3$ | $49.6 \pm 0.3$ |
| FT (Zero Head Init) (B.9.1) | $75.6 \pm 0.6$ | $59.9 \pm 0.5$ | $70.5 \pm 0.2$ | $82.2 \pm 0.6$ | $48.4 \pm 1.1$ |
| NoisyTune (B.9.2) | $73.4 \pm 0.3$ | $60.0 \pm 0.7$ | $70.3 \pm 0.2$ | $82.6 \pm 0.2$ | $48.9 \pm 0.5$ |
| Pretraining on Data Mix (B.9.3) + FT | $74.6 \pm 0.6$ | $60.1 \pm 0.6$ | $69.6 \pm 0.2$ | $81.6 \pm 0.4$ | $50.9 \pm 0.7$ |
| Pretext-Downstream Sum (B.9.4) + FT | $\underline{76.4 \pm 0.4}$ | $62.2 \pm 0.3$ | $67.1 \pm 0.1$ | $\underline{83.8 \pm 0.1}$ | $52.9 \pm 0.5$ |
| BiSSL (IJ Discarded) (B.9.5) + FT | $76.2 \pm 0.3$ | $\underline{63.0 \pm 0.3}$ | $\mathbf{71.7 \pm 0.1}$ | $81.5 \pm 0.3$ | $\underline{53.1 \pm 0.3}$ |
| BiSSL + FT | $\mathbf{78.3 \pm 0.2}$ | $\mathbf{64.2 \pm 0.4}$ | $\underline{71.5 \pm 0.1}$ | $\mathbf{84.1 \pm 0.2}$ | $\mathbf{55.7 \pm 0.2}$ |

## 4.4 Extended Evaluation

In this section, we present a series of experiments to examine how the performance of BiSSL compares with other baselines, and to explore how well BiSSL adapts to different configurations and experimental settings.

### 4.4.1 Comparative Baseline Evaluations

We assess BiSSL relative to several other baseline methods in this section. Due to computational constraints, we restrict evaluation to five downstream datasets: Pets, DTD, VOC07, Flowers, and CUB200. These cover both coarse- and fine-grained classification tasks across diverse and mostly disjoint visual domains. All baselines involving self-supervised pretraining use SimCLR as the pretext task, as described in Section 4.2.1. We briefly summarize the conceptual design of each baseline and defer full implementation details to Section B.9.

We include baselines consisting of a randomly initialized backbone, linear probing, and fine-tuning with a zero-initialized head, all described in Section B.9.1. Then, we further compare BiSSL to methods that similarly introduce an intermediate stage between pretraining and fine-tuning. These include NoisyTune (Wu et al., 2022), which perturbs backbone parameters with noise (details in Section B.9.2), continued pretraining on a mixture of downstream and pretext data (detailed in Section B.9.3), and joint minimization of a weighted sum of the downstream and pretext objectives (detailed in B.9.4). In addition, we also evaluate a BiSSL variant with the first term of the upper-level objective (2) discarded, eliminating the gradient coupling from the lower to upper-level, which effectively reduces to an alternating optimization scheme (details in B.9.5). This variant can itself also be considered an ablation, as it tests whether the increased information sharing imposed by the bilevel structure provides any tangible benefit for the downstream tasks.

Table 3 reports the results for all mentioned baselines, along with the FT and BiSSL+FT results from Table 1 for ease of comparison. As expected, pretrained backbones consistently outperform initializing them randomly, and fine-tuning yields further gains over linear probing. Notably, DTD and VOC07 exhibit the smallest relative drop in performance under linear probing compared to fine-tuning, suggesting that their tasks benefit more directly from the general-purpose representations captured during SSL pretraining. This could stem from the fact that VOC's labels are relatively coarse-grained, while DTD relies on general texture patterns rather than fine-grained distinctions, in contrast to the fine-grained Pets, Flowers, and CUB200 datasets. BiSSL overall achieves the strongest results, ranking first on four of the five datasets and second on the remaining one. In contrast, NoisyTune fails to improve downstream accuracy in this setting, and the remaining baselines provide gains over conventional fine-tuning less consistently, and when improvements do occur, they are generally smaller than those achieved by BiSSL.

Table 4: Downstream classification performance when using a ViT-backbone with MAE as the pretext task. BiSSL improves performance with statistical significance and reduces variance across all five datasets.

| **MAE** | Pets | DTD | VOC07 | Flowers | CUB200 |
|---|---|---|---|---|---|
| FT | $81.3 \pm 0.5$ | $61.3 \pm 0.7$ | $69.5 \pm 0.3$ | $81.1 \pm 1.1$ | $59.0 \pm 0.7$ |
| BiSSL+FT | $\mathbf{82.9 \pm 0.1}$ | $\mathbf{62.9 \pm 0.4}$ | $\mathbf{72.0 \pm 0.2}$ | $\mathbf{87.3 \pm 0.3}$ | $\mathbf{65.1 \pm 0.4}$ |
| *Avg Diff* | $\mathbf{+1.6}$ | $\mathbf{+1.6}$ | $\mathbf{+2.5}$ | $\mathbf{+6.2}$ | $\mathbf{+6.1}$ |

### 4.4.2 Transformer-Based Backbone Architecture

Our main experiments establish the effectiveness of BiSSL using ResNet-based architectures. To evaluate whether these benefits extend to other architectures, we test BiSSL with the more recent ViT backbone (Dosovitskiy et al., 2021), trained using the Masked Autoencoder (MAE) pretext task (He et al., 2022). Experimental details are provided in Section B.8. The evaluation uses the same five downstream datasets as in Section 4.4.1. As shown in Table 4, BiSSL delivers consistent and statistically significant improvements across all datasets and reduces variance in downstream accuracy compared to standard fine-tuning.

### 4.4.3 Varying the Pretraining Duration

To further assess the robustness of BiSSL, we vary the duration of self-supervised pretraining. Due to computational resource constraints, we adopt a smaller-scale version of the SimCLR setup, using a ResNet-18 backbone and the unlabeled partition of the smaller-scale STL10 (Coates et al., 2011) dataset for pretraining. Reusing the parameters from Section 4.2.2, BiSSL training corresponds to 100 conventional pretext epochs using the STL10 dataset. To ensure fair comparison, the pretext-only baselines are accordingly pretrained for 100 additional epochs. The rest of the setup remains identical to as described in Section 4.2. We evaluate on the flowers dataset, where BiSSL previously showed significant gains. Figure 2 depicts the final top-1 test accuracies achieved by separate models pretrained for varying durations. BiSSL consistently outperforms the baseline once sufficient pretraining is reached, consistent with the remarks in Section 3.4.

**Additional Results**  We provide additional results in Appendix C. Section C.1 reports training durations for self-supervised pretraining, BiSSL and fine-tuning. Section C.3 presents ablations on BiSSL by lowering the number of upper-level iterations to $N_U = 1$, underlining the benefits of increasing this value in practice. A sensitivity analysis of $\lambda$ is provided in Section C.4, which reports how classification accuracy is affected by varying this parameter. Section C.5 presents ImageNet fine-tuning results, and Section C.6 demonstrates that BiSSL also achieves significant performance improvements using the official SimCLR-pretrained weights from Chen et al. (2020) with the original $224 \times 244$ image resolution.

## 5 Exploratory Backbone Alignment Analyses

To assess the extent of which BiSSL improves backbone alignment for fine-tuning, we compare downstream alignment of self-supervised with BiSSL-trained backbones using quantitative and qualitative approaches.

### 5.1 Measuring Backbone Similarity

We first aim to quantitatively measure the backbone alignment. This is approached by calculating the cosine similarity (CS), firstly between backbone parameters and secondly between backbone output features. Specifically, we compare the CS between SimCLR-pretrained backbones and their fine-tuned versions, with the CS of BiSSL-trained backbones and their fine-tuned counterparts. CS values are computed for models trained on each of the 12 datasets listed in Table 1, with the average CS across these datasets reported in Figure 3. Further experimental setup details are provided in Section B.10.1, where we also provide individual

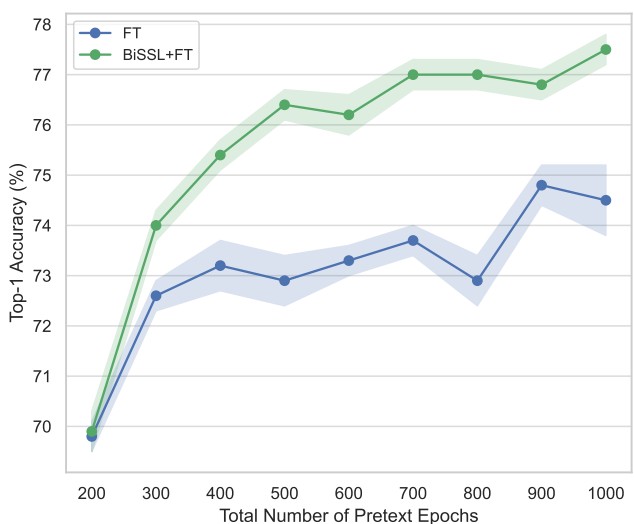

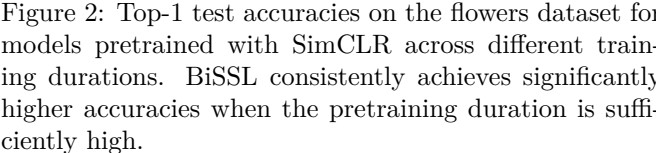

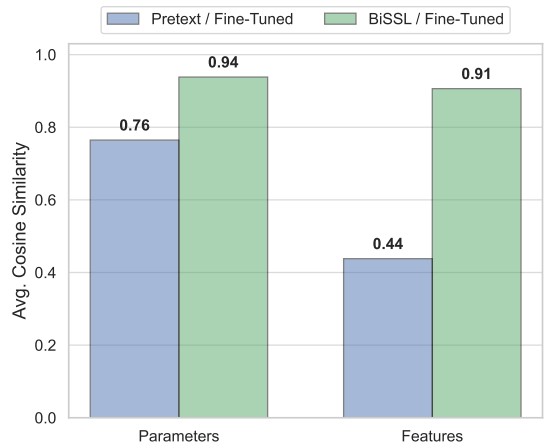

Figure 2: Top-1 test accuracies on the flowers dataset for models pretrained with SimCLR across different training durations. BiSSL consistently achieves significantly higher accuracies when the pretraining duration is sufficiently high.

Figure 3: Parameter-wise and feature-wise cosine similarities between SimCLR-pretrained and BiSSL-trained backbones with their respective fine-tuned counterparts. Results are averaged over the 12 downstream classification datasets listed in Table 1, and values for each individual dataset are listed in Table 9 of Section B.10.1. The results indicate that BiSSL increases downstream alignment of backbones.

dataset CS values in Table 9. The results imply that BiSSL backbones are consistently more aligned with their fine-tuned versions than standard SSL-pretrained backbones, implying that BiSSL-trained backbones impose a smaller parameter and feature shift during fine-tuning.

## 5.2 Visual Inspection of Latent Features

To better assess whether BiSSL nudges the latent features toward being more semantically meaningful for downstream tasks, we qualitatively inspect latent spaces using the t-Distributed Stochastic Neighbor Embedding (t-SNE) (Cieslak et al., 2020) method. Specifically, we compare features from SimCLR-pretrained backbones to those derived from BiSSL-trained lower-level backbones. Further implementation details are provided in Section B.10.2. Figure 4 illustrates the results on the flowers dataset, indicating that BiSSL improves downstream feature alignment. Additional plots on a selection of downstream datasets in Section C.7 reinforce this finding, demonstrating that this trend consistently persists.

## 6 Conclusion

This study introduces BiSSL, a novel training framework that integrates the self-supervised pretext and downstream fine-tuning objectives into a unified bilevel optimization problem. The bilevel structure enables the downstream objective to guide the pretext task optimization towards refining its representations to better align with the downstream task. We present a task-agnostic training algorithm and pipeline that incorporates BiSSL as a preparatory stage prior to fine-tuning. Experiments across multiple pretext and downstream tasks demonstrate that BiSSL consistently enhances fine-tuning performance, while additional alignment analyses indicate that it produces pretrained representations that better conform to downstream objectives. Overall, BiSSL demonstrates that explicitly addressing the misalignment between self-supervised pretrained backbones and downstream tasks prior to fine-tuning can yield substantial accuracy gains. This highlights the potential of bilevel optimization in this setting and motivates future research on training algorithms that explicitly optimize such alignment.

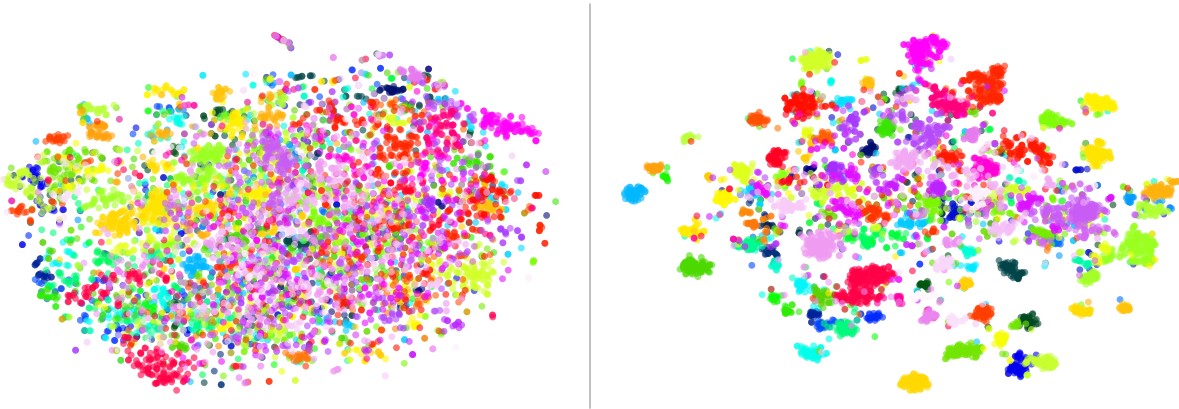

Figure 4: Visualizations of backbone output features on the Flowers dataset for the self-supervised pretrained backbone (left) and the BiSSL-trained backbone (right). Colors indicate different classes. Further details are provided in Section B.10.2, and analogous visualizations for other datasets are found in Section C.7. The BiSSL backbone output features exhibit improved discriminative alignment with the downstream classes.

## Broader Impact Statement

Our work focuses on improving downstream performance of self-supervised pretrained models without introducing new societal or ethical risks beyond those already associated with standard self-supervised learning and fine-tuning.

## Acknowledgments

This project is supported by the Pioneer Centre for Artificial Intelligence, Denmark.[1] We acknowledge the Danish e-Infrastructure Consortium (DeiC) for awarding this project access to the LUMI supercomputer. Lastly, the authors would like to thank Sijia Liu and Yihua Zhang (Michigan State University) for providing valuable feedback in a discussion, which helped to refine and solidify our perspective on the topic of integrating bilevel optimization in deep learning.

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

# A  Theoretical Insights and Framework Comparisons in BiSSL

## A.1  Derivation of the Implicit Jacobian

Assume the setup of the BiSSL optimization problem described in (2) and (3). In the following derivations, we will assume that $\boldsymbol{\phi}_P$ is fixed, allowing us to simplify the expressions involved. To streamline the notation further, we continue to use the convention $\nabla_{\boldsymbol{\xi}} h(\boldsymbol{\xi})|_{\boldsymbol{\xi}=\boldsymbol{\psi}} := \nabla_{\boldsymbol{\xi}} h(\boldsymbol{\psi})$, when it is clear from context which variables are differentiated with respect to. Under these circumstances, we then define the lower-level objective from (3) as

$$G(\boldsymbol{\theta}_D, \boldsymbol{\theta}_P) := \mathcal{L}^P(\boldsymbol{\theta}_P, \boldsymbol{\phi}_P) + \lambda r(\boldsymbol{\theta}_P, \boldsymbol{\theta}_D). \tag{8}$$

Now, furthermore assume that the regularization objective $r$ is strongly convex, and that $\mathcal{L}^P$ is twice-differentiable and $L$-smooth (i.e. it has Lipschitz-continuous gradients). As established and similarly exploited in prior works (Rajeswaran et al., 2019; Zhang et al., 2022b; 2023a), these properties imply that an adequate scaling of $\lambda$ renders the lower-level objective $G$ effectively convex. This is advantageous because assuming convexity of $G$ ensures that for any $\boldsymbol{\theta}_D \in \mathbb{R}^L$, there exists a corresponding $\hat{\boldsymbol{\theta}}_P \in \mathbb{R}^L$ that satisfies the stationary condition $\nabla_{\boldsymbol{\theta}_P} G(\boldsymbol{\theta}_D, \hat{\boldsymbol{\theta}}_P) = \mathbf{0}$. In other words, we are assured that a potential minimizer of $G(\boldsymbol{\theta}_D, \cdot)$ exists for all $\boldsymbol{\theta}_D \in \mathbb{R}^L$. Now, further assume that the Hessian matrix $\nabla_{\boldsymbol{\theta}_P}^2 G(\boldsymbol{\theta}_D, \hat{\boldsymbol{\theta}}_P)$ is invertible for all $\boldsymbol{\theta}_D \in \mathbb{R}^L$. Under these conditions, the implicit function theorem (Dontchev & Rockafellar, 2014; Zucchet & Sacramento, 2022) guarantees the existence of an implicit unique and *differentiable* function $\boldsymbol{\theta}_P^* : \mathcal{N}(\boldsymbol{\theta}_D) \to \mathbb{R}^L$, with $\mathcal{N}(\boldsymbol{\theta}_D)$ being a neighborhood of $\boldsymbol{\theta}_D$, that satisfies $\boldsymbol{\theta}_P^*(\boldsymbol{\theta}_D) = \hat{\boldsymbol{\theta}}_P$ and $\nabla_{\boldsymbol{\theta}_P} G(\tilde{\boldsymbol{\theta}}_D, \boldsymbol{\theta}_P^*(\tilde{\boldsymbol{\theta}}_D)) = \mathbf{0}$ for all $\tilde{\boldsymbol{\theta}}_D \in \mathcal{N}(\boldsymbol{\theta}_D)$.

As we then conclude that the lower-level solution $\boldsymbol{\theta}_P^*(\boldsymbol{\theta}_D)$ is indeed a differentiable function under these conditions, this justifies that the expression

$$\frac{\mathrm{d}}{\mathrm{d}\boldsymbol{\theta}_D}[\nabla_{\boldsymbol{\theta}_P} G(\boldsymbol{\theta}_D, \boldsymbol{\theta}_P^*(\boldsymbol{\theta}_D))] = \mathbf{0}$$

is well-posed for all $\boldsymbol{\theta}_D \in \mathbb{R}^L$. By applying the chain rule, the expression becomes

$$\nabla_{\boldsymbol{\theta}_D \boldsymbol{\theta}_P}^2 G(\boldsymbol{\theta}_D, \boldsymbol{\theta}_P^*(\boldsymbol{\theta}_D)) + \frac{\mathrm{d}\boldsymbol{\theta}_P^*(\boldsymbol{\theta}_D)}{\mathrm{d}\boldsymbol{\theta}_D}^T \nabla_{\boldsymbol{\theta}_P}^2 G(\boldsymbol{\theta}_D, \boldsymbol{\theta}_P^*(\boldsymbol{\theta}_D)) = \mathbf{0}.$$

Recalling that $\nabla_{\boldsymbol{\theta}_P}^2 G(\boldsymbol{\theta}_D, \boldsymbol{\theta}_P^*(\boldsymbol{\theta}_D))$ is assumed invertible, the implicit Jacobian (IJ) $\frac{\mathrm{d}\boldsymbol{\theta}_P^*(\boldsymbol{\theta}_D)}{\mathrm{d}\boldsymbol{\theta}_D}^T$ can be isolated

$$\frac{\mathrm{d}\boldsymbol{\theta}_P^*(\boldsymbol{\theta}_D)}{\mathrm{d}\boldsymbol{\theta}_D}^T = -\nabla_{\boldsymbol{\theta}_D \boldsymbol{\theta}_P}^2 G(\boldsymbol{\theta}_D, \boldsymbol{\theta}_P^*(\boldsymbol{\theta}_D)) \big[\nabla_{\boldsymbol{\theta}_P}^2 G(\boldsymbol{\theta}_D, \boldsymbol{\theta}_P^*(\boldsymbol{\theta}_D))\big]^{-1},$$

and by substituting the expression for $G$ from (8), the expression becomes

$$\frac{\mathrm{d}\boldsymbol{\theta}_P^*(\boldsymbol{\theta}_D)}{\mathrm{d}\boldsymbol{\theta}_D}^T = -\lambda \nabla_{\boldsymbol{\theta}_D \boldsymbol{\theta}_P}^2 r(\boldsymbol{\theta}_P^*(\boldsymbol{\theta}_D), \boldsymbol{\theta}_D) \big[\nabla_{\boldsymbol{\theta}_P}^2 \big(\mathcal{L}^P(\boldsymbol{\theta}_P^*(\boldsymbol{\theta}_D), \boldsymbol{\phi}_P) + \lambda r(\boldsymbol{\theta}_P^*(\boldsymbol{\theta}_D), \boldsymbol{\theta}_D)\big)\big]^{-1}$$

$$= -\nabla_{\boldsymbol{\theta}_D \boldsymbol{\theta}_P}^2 r(\boldsymbol{\theta}_P^*(\boldsymbol{\theta}_D), \boldsymbol{\theta}_D) \Big[\nabla_{\boldsymbol{\theta}_P}^2 \Big(\frac{1}{\lambda}\mathcal{L}^P(\boldsymbol{\theta}_P^*(\boldsymbol{\theta}_D), \boldsymbol{\phi}_P) + r(\boldsymbol{\theta}_P^*(\boldsymbol{\theta}_D), \boldsymbol{\theta}_D)\Big)\Big]^{-1}. \tag{9}$$

**To summarize**, given the following assumptions:

- The lower-level pretext head parameters $\boldsymbol{\phi}_P$ are fixed.

- The regularization objective $r$ is strongly convex.

- The objective $\mathcal{L}^P$ is twice-differentiable and $L$-smooth, with $\lambda$ accordingly chosen so that $G$ is convex.

- The Hessian matrix $\nabla_{\boldsymbol{\theta}_P}^2 G(\boldsymbol{\theta}_D, \boldsymbol{\theta}_P^*(\boldsymbol{\theta}_D))$ is invertible for all $\boldsymbol{\theta}_D \in \mathbb{R}^L$.

Then, the IJ $\frac{\mathrm{d}\boldsymbol{\theta}_P^*(\boldsymbol{\theta}_D)}{\mathrm{d}\boldsymbol{\theta}_D}^T$ can be explicitly expressed by (9). The authors further acknowledge that an explicit expression for the IJ without fixing $\boldsymbol{\phi}_P$ is achievable, though this is left for future exploration. The assumptions in the final two bullets above may not strictly hold in practice with deep neural networks, yet the resulting gradient expressions have nonetheless proven practically beneficial both in this and prior work.

## A.2 Distinction from Bilevel Optimization in Meta-Learning

While bilevel optimization (BLO) has been applied in meta-learning frameworks such as MAML (Finn et al., 2017), Sign-MAML (Fan et al., 2021) and iMAML (Rajeswaran et al., 2019), BiSSL represents a distinct application and implementation of BLO, tailored for the challenges of self-supervised learning (SSL). In the aforementioned works, BLO is primarily utilized to address few-shot learning scenarios, focusing on efficiently adapting models to new tasks with minimal labeled data. Conversely, BiSSL applies BLO to concurrently manage the more complex task of self-supervised pretext pretraining on unlabeled data with downstream fine-tuning on labeled data. Another key distinction is that in meta-learning, the upper- and lower-level objectives are closely related, with the upper-level objective formulated as a summation of the lower-level tasks. In contrast, BiSSL involves fundamentally distinct objectives at each level, utilizing separate datasets and tasks for pretraining and fine-tuning. This design allows BiSSL to better align the pretrained model with the requirements of a specific downstream task. Conversely, the BLO in meta-learning aims to broadly generalize across a wide range of tasks, prioritizing adaptability rather than task-specific optimization.

# B Experimental Details

## B.1 Dataset Partitions

The Caltech-101 (Li et al., 2022a) dataset does not come with a pre-defined train/test split, so the same convention as previous works is followed (Chen et al., 2020; Donahue et al., 2014; Simonyan & Zisserman, 2014), where 30 random images per class are selected for the training partition, and the remaining images are assigned for the test partition. For the DTD (Cimpoi et al., 2014) and SUN397 (Xiao et al., 2010) datasets, which offer multiple proposed train/test partitions, the first splits are used, consistent with the approach in (Chen et al., 2020).

For downstream hyperparameter optimization, portions of the training partitions from each respective labeled dataset are designated as validation datasets. The FGVC Aircraft (Maji et al., 2013), Oxford 102 Flowers (Nilsback & Zisserman, 2008), DTD, and Pascal VOC 2007 (Everingham et al., 2010) datasets already have designated validation partitions. For the remaining labeled datasets, we reserve roughly 20% of the training data for validation. For CityScapes (Cordts et al., 2016), this split is applied separately within each city, while for the other datasets, validation partitions are randomly sampled while ensuring that the class proportions are preserved.

## B.2 Downstream Task Fine-Tuning of the Baseline Setup

A random grid search of 100 hyper-parameter configurations for the base learning rates and weight decays is conducted, where one model is fine-tuned for each configuration. Base learning rates and weight decays are log-uniformly sampled over the ranges of $10^{-4}$ to 1 and $10^{-5}$ to $10^{-2}$, respectively. Validation data accuracy is evaluated after each epoch. The hyper-parameter configuration yielding the best balance between high validation accuracy and low validation loss is considered the optimal hyper-parameter configuration.[2] The corresponding optimal hyper-parameters for each downstream dataset are documented in Table 5. For subsequent evaluation on the test data, we train 10 models with different random seeds, each using the considered optimal hyper-parameter configurations. During the training of each respective model, the model parameters are stored after each epoch if the top-1 validation accuracy (or 11-point mAP for the VOC07 dataset) has increased compared to the previous highest top-1 validation accuracy achieved during training. Top-1 and top-5 test data accuracies (or 11-point mAP for the VOC07 dataset) are evaluated for each of the 10 models, from which the calculated means and standard deviations of these accuracies are documented.

---

[2]In certain scenarios during the experiments, the configuration that achieved the highest validation accuracy also yielded a notably higher relative validation loss. To ensure better generalizability, an alternative configuration with a more favorable trade-off was selected in these cases.

Table 5: Optimal hyper-parameter configurations used for downstream fine-tuning after conventional pretext pretraining.

| Dataset | SimCLR | | BYOL | |
|---|---|---|---|---|
| | Learning Rate | Weight Decay | Learning Rate | Weight Decay |
| Food | 0.0167 | 0.00613 | 0.0513 | 0.00147 |
| CIFAR10 | 0.0033 | 0.00158 | 0.0014 | 0.00106 |
| CIFAR100 | 0.0027 | 0.00012 | 0.0023 | 0.00012 |
| CUB200 | 0.0409 | 0.0084 | 0.0095 | 0.00594 |
| SUN397 | 0.0069 | 0.00003 | 0.004 | 0.00003 |
| Cars | 0.0377 | 0.00454 | 0.115 | 0.00257 |
| Aircrafts | 0.0269 | 0.0038 | 0.0119 | 0.00333 |
| VOC07 | 0.0054 | 0.0089 | 0.0032 | 0.00616 |
| DTD | 0.0514 | 0.0011 | 0.0114 | 0.00011 |
| Pets | 0.0378 | 0.00114 | 0.0044 | 0.00779 |
| Caltech101 | 0.0131 | 0.00005 | 0.0069 | 0.00027 |
| Flowers | 0.2178 | 0.00046 | 0.035 | 0.00262 |

## B.3 Downstream Head Warmup and Upper-level of BiSSL

Table 6 outlines the learning rates and weight decays used for the downstream head warm-up and upper-level of BiSSL of each respective downstream dataset, as described in the BiSSL experimental setup of Section 4.2.2. The first term of the upper-level gradient (7) is approximated using the Conjugate Gradient

Table 6: Hyper-parameters used for the Downstream Head Warm-up and Upper-level of BiSSL.

| Dataset | SimCLR | | BYOL | |
|---|---|---|---|---|
| | Learning Rate | Weight Decay | Learning Rate | Weight Decay |
| Food | 0.03 | 0.001 | 0.035 | 0.002 |
| CIFAR10 | 0.015 | 0.001 | 0.01 | 0.001 |
| CIFAR100 | 0.01 | 0.0001 | 0.01 | 0.001 |
| CUB200 | 0.03 | 0.001 | 0.015 | 0.0001 |
| SUN397 | 0.015 | 0.00005 | 0.01 | 0.00005 |
| Cars | 0.035 | 0.001 | 0.04 | 0.002 |
| Aircrafts | 0.03 | 0.005 | 0.015 | 0.003 |
| VOC07 | 0.015 | 0.005 | 0.005 | 0.006 |
| DTD | 0.03 | 0.001 | 0.015 | 0.0001 |
| Pets | 0.03 | 0.001 | 0.02 | 0.002 |
| Caltech101 | 0.03 | 0.0001 | 0.015 | 0.0002 |
| Flowers | 0.05 | 0.0005 | 0.035 | 0.002 |

(CG) method (Nazareth, 2009; Shewchuk, 1994). Our implementation follows a similar structure to that used in Rajeswaran et al. (2019), employing $N_c = 5$ iterations and a dampening term $\lambda_{\mathrm{damp}} = 10$. Given matrix $A$ and vector $\mathbf{v}$, the CG method iteratively approximates $A^{-1}\mathbf{v}$, which requires evaluation of multiple matrix-vector products $A\mathbf{d}_1$, ..., $A\mathbf{d}_{N_c}$. In practice, storing the matrix $A$ (in our case, the Hessian $\nabla^2_{\boldsymbol{\theta}_P}\mathcal{L}^P(\boldsymbol{\theta}^*_P(\boldsymbol{\theta}_D), \boldsymbol{\phi}_P))$ in its full form is often infeasible. Instead, a function that efficiently computes the required matrix-vector products instead of explicitly storing the matrix is typically utilized. For transparency, the function employed in our setup is detailed in Algorithm 2. This approach ensures that the output of the CG algorithm is an approximation of the inverse Hessian-vector product in the first term of Equation (7) as intended.

---

**Algorithm 2** Hessian Vector Product Calculation $f_H$ (To use in the CG Algorithm)

---

1: **Input:** Input vector $\mathbf{v}$. Model parameters $\boldsymbol{\theta}_P$, $\boldsymbol{\phi}_P$. Training objective $\mathcal{L}^P$. Lower-level data batch $\mathbf{x}$. Regularization weight $\lambda$ and dampening $\lambda_{\mathrm{damp}}$.

2: $\pi(\boldsymbol{\theta}_P) \leftarrow \left(\nabla_{\boldsymbol{\theta}}\mathcal{L}^P(\boldsymbol{\theta}, \boldsymbol{\phi}_P; \mathbf{x})\big|_{\boldsymbol{\theta}=\boldsymbol{\theta}_P}\right)^T\mathbf{v}$

3: $\mathbf{g} \leftarrow \nabla_{\boldsymbol{\theta}}\pi(\boldsymbol{\theta})\big|_{\boldsymbol{\theta}=\boldsymbol{\theta}_P}$         ▷ Memory efficient calculation of $\nabla^2_{\boldsymbol{\theta}}\mathcal{L}^P(\boldsymbol{\theta}, \boldsymbol{\phi}_P; \mathbf{x})\big|_{\boldsymbol{\theta}=\boldsymbol{\theta}_P}\mathbf{v}$.

4: $\mathbf{y} \leftarrow \mathbf{v} + \frac{1}{\lambda+\lambda_{\mathrm{damp}}}\mathbf{y}$

5: **Return:** $f_H(\mathbf{v}) := \mathbf{y}$

---

## B.4 Composite Configuration of BiSSL

To avoid data being reshuffled between every training stage alternation, the respective batched lower- and upper-level training datasets are stored in separate stacks from which data is drawn. The stacks are only "reset" when the number of remaining batches is smaller than the number of gradient steps required before alternating to the other level. For example, the lower-level stack is reshuffled every fourth training stage alternation. If the downstream dataset does not provide enough data for making $N_U = 8$ batches with non-overlapping data points, the data is simply reshuffled every time the remaining number of data points is smaller than the upper-level batch size (e.g. 256 images in the classification experiments).

## B.5 Downstream Fine-Tuning after BiSSL

The learning rates and weight decays used for downstream fine-tuning after BiSSL for each respective downstream dataset are outlined in Table 7. Section 4.2.2 outlines the experimental setup.

## B.6 Statistical Significance Tests

As described in Section B.2, each documented accuracy is obtained by fine-tuning 10 models with different random seeds. When comparing two sets of observations, such as FT and BiSSL+FT in Table 1, this yields two groups of 10 values each, for a total of 20 observations. Under the standard assumption that the values within each group are independent and identically distributed, we use a two-sample permutation test to assess whether the average accuracy of each group differs significantly.

Under the null hypothesis that all 20 values come from the same distribution, the test pools the values, randomly assigns 10 to each group, and computes the resulting difference between group averages for each permutation. Since there are $\binom{20}{10} = 184,756$ unique permutations, we perform the exact test by evaluating all possible assignments.

The resulting p-value is the proportion of permutations where the difference of averages is at least as large as the observed difference. We consider p-values below 0.01 to indicate a statistically significant difference in average accuracy between the two groups.

Table 7: Optimal hyper-parameter configurations used for downstream fine-tuning after BiSSL.

| Dataset | SimCLR | | BYOL | |
|---|---|---|---|---|
| | Learning Rate | Weight Decay | Learning Rate | Weight Decay |
| Food | 0.00059 | 0.00008 | 0.00018 | 0.00019 |
| CIFAR10 | 0.00097 | 0.00258 | 0.00098 | 0.00039 |
| CIFAR100 | 0.00252 | 0.00046 | 0.00042 | 0.00292 |
| CUB200 | 0.00043 | 0.00251 | 0.00066 | 0.00075 |
| SUN397 | 0.00079 | 0.00001 | 0.00063 | 0.00029 |
| Cars | 0.05706 | 0.00412 | 0.06221 | 0.00317 |
| Aircrafts | 0.00015 | 0.00003 | 0.00025 | 0.00056 |
| VOC07 | 0.00022 | 0.00592 | 0.00029 | 0.004 |
| DTD | 0.00062 | 0.00002 | 0.00028 | 0.00008 |
| Pets | 0.00153 | 0.00314 | 0.00042 | 0.00002 |
| Caltech101 | 0.00079 | 0.00005 | 0.00194 | 0.00003 |
| Flowers | 0.01768 | 0.0009 | 0.00058 | 0.00006 |

### B.7 Object Detection and Semantic Segmentation

The experimental setup primarily follows the main implementation specified in Section 4.2, with the specific modifications made specified here.

**Object Detection** We utilize a Faster R-CNN with a ResNet-50 C4 backbone (Ren et al., 2015) as the downstream model architecture. In line with Grill et al. (2020); He et al. (2020); Bardes et al. (2022), we assign the box prediction head as the $conv_5$ stage of the ResNet-50 followed by global pooling. We use anchors of size 32, 64, 128 and 256 with aspect ratios 0.5, 1.0 and 2.0. The downstream data augmentations involve rescaling the images so that their longest edge is between 196 and 320 pixels, followed by normalization. Downstream fine-tuning is conducted for 50 epochs with a batch size of 16. The random hyperparameter grid search is performed across 50 distinct configurations of learning rates and weight decays, stochastically sampled log-uniformly within the intervals $10^{-4}$ to $10^{-1}$ and $10^{-6}$ to $10^{-2}$, respectively. The linear head warm-up, conducted prior to BiSSL, is executed for 5 epochs.

We note that, since the downstream head is inserted between intermediate stages of the backbone, this setup does not fully comply with the notation in Section 3.1, where the downstream head is assumed to operate solely on the backbone's final output layer. Nonetheless, the derived gradient expression (7) retains its validity in this setting, as it does not depend on any part of the downstream head but only requires a one-to-one correspondence between the parameter-wise components between the levels. In particular, the $conv_5$ stage parameters of the lower-level backbone can still be directly substituted into the corresponding $conv_5$ stage of the upper-level backbone.

**Semantic Segmentation** For semantic segmentation tasks, we adopt an FCN-based architecture (Long et al., 2015a) similar to the setup used in Grill et al. (2020); He et al. (2020). We use the ResNet-50 backbone output prior to global pooling, and the $3 \times 3$ convolutions in the $conv_5$ block are reassigned dilation 2 and stride 1. The downstream head is composed of two $3 \times 3$ convolutions, each with dilation 6 and 256 channels, with batch normalization and ReLU activation applied in between. A final $1 \times 1$ convolution performs the per-pixel classification required for segmentation, resulting in a total stride of 16 (corresponding to the FCN-16 setup (Long et al., 2015a)).

Data augmentation includes resizing such that the shorter image side is 192 pixels, followed by random horizontal flipping with probability 0.5 and normalization. The fine-tuning hyperparameter search is identical to the object detection setup, and the linear head warm-up is similarly conducted for 5 epochs. Although some structural modifications are applied to the backbone, similar to the object detection configuration, these changes do not introduce any compatibility issues with BiSSL.

### B.8 Masked Autoencoder with ViT Backbone

Due to computational constraints, we adopt a relatively lightweight configuration with a ViT-S backbone (Touvron et al., 2021; Heo et al., 2021) and an $8 \times 8$ patch size. The decoder matches the original MAE implementation (He et al., 2022). Pretext pretraining was run for 500 epochs using the AdamW (Loshchilov & Hutter, 2019) optimizer with a cosine-decayed learning rate starting at 0.0005, weight decay of 0.05, and momentum coefficients of $\beta_1 = 0.9$ and $\beta_2 = 0.95$. The lower-level BiSSL stage used the same optimizer but with an initial learning rate of 0.0001. For both upper-level optimization and downstream fine-tuning, we applied a 10% drop path rate and layer-wise learning rate decay with a factor of 0.75. We conducted the random hyperparameter grid search over 50 different configurations. All other experimental settings match those described in Section 4.2.2.

### B.9 Additional Baseline Comparisons

This section outlines the complete implementation details for each newly added baseline according the experiments summarized in Table 3 of Section 4.4.1. In all experiments within this section involving self-supervised pretraining, we use SimCLR as the pretext task.

#### B.9.1 Random Initialization, Linear Probing and Zero-Initialized Classification Head

The first baseline uses randomly initialized backbone weights instead of the SimCLR-pretrained backbone. The linear probing baseline freezes the backbone weights, and the zero-initialized head baseline sets the classification head parameters to zero rather than random values. As prior work suggests that randomly initialized heads may cause a distortion of pretrained features during fine-tuning (Kumar et al., 2022; Yang et al., 2022), we hypothesize that zero-initializing the downstream head may mitigate such loss. For each result, the hyperparameter grid search is conducted over 25 different combinations. All other aspects of the experimental setup follow Section 4.2.1. Results are included in the respective "Random Initialization, "Linear Probing" and "FT (Zero Head Init)" rows of Table 3.

#### B.9.2 NoisyTune

NoisyTune (Wu et al., 2022) applies layer-wise Gaussian noise to the backbone weights prior to fine-tuning, with the noise magnitude determined by each layer's statistics. We use a noise scale of 0.015, and conduct the hyperparameter grid search over 50 different combinations. All other aspects of fine-tuning follow Section 4.2.1. Results are reported in the "NoisyTune" row of Table 3.

#### B.9.3 Continued Pre-Training on Downstream and Pretext Dataset Mixture

For each downstream dataset, we resumed self-supervised pretraining after adding the downstream data to the original pretraining data pool. This baseline thus continues pretraining on a mixture of pretext and downstream data before fine-tuning.

Given the current BiSSL setting conducts what is roughly equivalent to 8 pretext epochs, we here continued SSL pretraining for 10 epochs using a cosine-decayed learning rate starting at 1.0. All other pretraining settings match those described in Section 4.2.1. The subsequent fine-tuning follows the standard setup, but with a hyperparameter grid search over 25 combinations. Results for this baseline are reported in the "Pretraining on Data Mix + FT" row of Table 3.

Table 8: Top-1 and Top-5 accuracies on the DTD dataset using different values of $w$ for the weighted objective baseline in (10), compared with conventional fine-tuning and BiSSL. Performance peaks at $w = 0.25$, but still lags behind BiSSL.

| Accuracy | $w = 0$ | $w = 0.05$ | $w = 0.1$ | $w = 0.25$ | $w = 0.5$ | $w = 0.75$ | FT | BiSSL+FT |
|---|---|---|---|---|---|---|---|---|
| Top-1 | $60.1 \pm 0.3$ | $61.6 \pm 0.2$ | $61.7 \pm 0.2$ | $\underline{62.2 \pm 0.3}$ | $60.8 \pm 0.3$ | $60.3 \pm 0.5$ | $60.3 \pm 0.9$ | $\mathbf{64.2 \pm 0.4}$ |
| Top-5 | $\underline{86.3 \pm 0.3}$ | $86.2 \pm 0.3$ | $86.2 \pm 0.3$ | $86.1 \pm 0.3$ | $86.2 \pm 0.3$ | $85.7 \pm 0.3$ | $85.8 \pm 0.6$ | $\mathbf{87.8 \pm 0.3}$ |

### B.9.4 Weighted Sum of Pretext and Downstream Objectives

This baseline is trained by solving the single-level optimization problem:

$$\min_{\boldsymbol{\theta}, \boldsymbol{\phi}_P, \boldsymbol{\phi}_D} (1 - w)\mathcal{L}^P(\boldsymbol{\theta}, \boldsymbol{\phi}_P; \mathcal{D}^P) + w\mathcal{L}^D(\boldsymbol{\theta}, \boldsymbol{\phi}_D; \mathcal{D}^D), \tag{10}$$

where $w \in [0, 1]$ controls the relative weighting of the pretext and downstream objectives. While this method offers a more straightforward alternative to BiSSL, it lacks the principled bilevel optimization structure, and as our results will show, its performance improvements are less effective.

We use the lower-level optimizer configuration from BiSSL (see Section 4.2.2) and train for 4000 steps, matching the total number of upper-level iterations in the default BiSSL configuration. Fine-tuning follows the main setup but with a grid search over 50 hyperparameter combinations. Backbone gradients for the two loss terms are computed separately on their respective distinct mini-batches, scaled by $1 - w$ and $w$ respectively, and then summed prior to the update step.

To determine the suitable size of $w$, we conducted a sweep on the DTD dataset over different values of $w$ in the range between 0 and 1, evaluating both top-1 and top-5 classification performance. For each value of $w$, we conduct the subsequent fine-tuning hyperparameter grid search over 25 different combinations, otherwise the fine-tuning setup remains identical to as described in Section 4.2.1. The results of the sweep are shown in Table B.9.3.

The results imply that $w = 0.25$ provides the best top-1 classification accuracy. In contrast to BiSSL, the top-5 accuracy does generally not show a significant increase compared to conventional fine-tuning, regardless of the value of $w$. Setting $w = 0$ essentially reduces the training to pretext-only learning, reflected by comparable performance to conventional fine-tuning (the "FT" column), with slightly lower top-1 accuracy and slightly higher top-5 accuracy. Based on the results, we consider $w = 0.25$ as the optimal value for this baseline.

Using $w = 0.25$, we summarize the baseline experiments in the "Pretext-Downstream Sum + FT" row in Table 3.

### B.9.5 Discarded First Term of Upper-Level Objective in BiSSL

For this baseline, we discard the first term of the upper-level objective in (2), which contains the IJ. This blocks the propagation of lower-level gradient information to the upper-level, hence instead resembling a more conventional alternating optimization scheme where only parameter values are shared. The rest of the setup follows the default configuration in Section 4.2.2, except that the fine-tuning hyperparameter grid search was conducted for 50 different combinations. The results are provided in the "BiSSL (IJ Discarded) + FT"-row of Table 3.

Table 9: Parameter-wise and feature-wise cosine similarities (multiplied by 100 for better readability) between SimCLR-pretrained and BiSSL-trained backbones with their respective fine-tuned counterparts for each dataset. In every case, the BiSSL/Fine-Tuned cosine similarities are statistically significantly higher, as indicated by bold font.

| Dataset | Cosine Similarity (Parameters) | | Cosine Similarity (Features) | |
|---|---|---|---|---|
| | Pretext/Fine-Tuned | BiSSL/Fine-Tuned | Pretext/Fine-Tuned | BiSSL/Fine-Tuned |
| Food | $15.88 \pm 0.22$ | $\mathbf{99.79 \pm 0.03}$ | $16.77 \pm 0.43$ | $\mathbf{93.50 \pm 0.37}$ |
| CIFAR10 | $99.01 \pm 0.46$ | $\mathbf{99.84 \pm 0.01}$ | $40.02 \pm 2.65$ | $\mathbf{80.66 \pm 0.42}$ |
| CIFAR100 | $99.50 \pm 0.03$ | $\mathbf{99.69 \pm 0.07}$ | $56.00 \pm 0.46$ | $\mathbf{89.75 \pm 0.97}$ |
| CUB200 | $17.01 \pm 0.20$ | $\mathbf{99.99 \pm 0.00}$ | $26.74 \pm 0.26$ | $\mathbf{96.28 \pm 0.02}$ |
| SUN397 | $99.45 \pm 0.09$ | $\mathbf{99.97 \pm 0.00}$ | $58.33 \pm 0.28$ | $\mathbf{97.11 \pm 0.08}$ |
| Cars | $24.42 \pm 0.11$ | $\mathbf{26.64 \pm 0.10}$ | $28.78 \pm 0.34$ | $\mathbf{65.01 \pm 0.14}$ |
| Aircrafts | $65.74 \pm 0.15$ | $\mathbf{99.95 \pm 0.01}$ | $31.42 \pm 0.27$ | $\mathbf{90.97 \pm 0.24}$ |
| Pets | $98.87 \pm 0.27$ | $\mathbf{99.99 \pm 0.00}$ | $45.28 \pm 1.64$ | $\mathbf{95.68 \pm 0.26}$ |
| DTD | $99.26 \pm 0.01$ | $\mathbf{99.99 \pm 0.00}$ | $47.83 \pm 0.23$ | $\mathbf{94.40 \pm 0.27}$ |
| VOC07 | $99.91 \pm 0.01$ | $\mathbf{99.99 \pm 0.00}$ | $58.27 \pm 0.93$ | $\mathbf{94.04 \pm 0.07}$ |
| Caltech101 | $99.86 \pm 0.01$ | $\mathbf{99.99 \pm 0.00}$ | $67.97 \pm 0.37$ | $\mathbf{97.64 \pm 0.08}$ |
| Flowers | $98.61 \pm 0.02$ | $\mathbf{99.98 \pm 0.01}$ | $48.22 \pm 0.11$ | $\mathbf{92.46 \pm 0.43}$ |
| *Average* (Fig. 3) | 76.46 | **93.82** | 43.80 | **90.63** |

## B.10 Backbone Alignment Analysis

### B.10.1 Cosine Similarities

Recall the notation in Section 3.1, where we let $\boldsymbol{\theta}$ denote the backbone parameters obtained after pretext task training, and $\boldsymbol{\theta}_P^*(\boldsymbol{\theta}_D)$ the backbone after applying BiSSL. We now further define $\boldsymbol{\theta}_i^{FT}$ and $\boldsymbol{\theta}_P^*(\boldsymbol{\theta}_D)_i^{FT}$ for $i = 1, \ldots, 10$ as the fine-tuned backbone parameters of the pretext and BiSSL models, respectively, which we achieve 10 of for each downstream dataset (recall Section B.2). Lastly, let

$$cs_N(\mathbf{x}, \mathbf{y}) = \frac{\mathbf{x}^T \mathbf{y}}{\|\mathbf{x}\|_2^2 \|\mathbf{y}\|_2^2}, \quad \mathbf{x}, \mathbf{y} \in \mathbb{R}^N / \{\mathbf{0}\} \tag{11}$$

denote the cosine similarity of $N$-dimensional vectors.

The purpose of these evaluations is then to measure how aligned the pretext pretrained backbones and their fine-tuned counterparts $\boldsymbol{\theta}_P^*(\boldsymbol{\theta}_D)$ and $\boldsymbol{\theta}_i^{FT}$ are compared to the BiSSL-trained backbones and their fine-tuned counterparts $\boldsymbol{\theta}_P^*(\boldsymbol{\theta}_D)$ and $\boldsymbol{\theta}_P^*(\boldsymbol{\theta}_D)_i^{FT}$. Specifically, we first compute the parameter-wise cosine similarities

$$cs_L(\boldsymbol{\theta}, \boldsymbol{\theta}_i^{FT}) \qquad \text{and} \qquad cs_L(\boldsymbol{\theta}_P^*(\boldsymbol{\theta}_D), \boldsymbol{\theta}_P^*(\boldsymbol{\theta}_D)_i^{FT}), \tag{12}$$

for $i = 1, \ldots, 10$. For each downstream dataset, the mean and standard deviation of these values are reported in the left column of Table 9.

We also compute the average cosine similarity between output features of the respective backbones for each downstream dataset on its test partition $\mathcal{D}^{D_{\text{test}}} = \{\mathbf{x}_k, \mathbf{y}_k\}_{k=0}^{|\mathcal{D}^{D_{\text{test}}}|}$, given by

$$\frac{1}{|\mathcal{D}^{D_{\text{test}}}|} \sum_{\mathbf{x}, \mathbf{y} \in \mathcal{D}^{D_{\text{test}}}} cs_M(f_{\boldsymbol{\theta}}(\mathbf{x}), f_{\boldsymbol{\theta}_i^{FT}}(\mathbf{x})) \qquad \text{and} \qquad \frac{1}{|\mathcal{D}^{D_{\text{test}}}|} \sum_{\mathbf{x}, \mathbf{y} \in \mathcal{D}^{D_{\text{test}}}} cs_M(f_{\boldsymbol{\theta}_P^*(\boldsymbol{\theta}_D)}(\mathbf{x}), f_{\boldsymbol{\theta}_P^*(\boldsymbol{\theta}_D)_i^{FT}}(\mathbf{x})), \tag{13}$$

Table 10: Total GPU hours required for pretext pretraining and BiSSL training with the SimCLR pretext task. BiSSL introduces some overhead relative to fine-tuning, while its computational cost is only a fraction of that required for pretext pretraining.

| | **Computation Time** |
|---|---|
| Pretext Training | 1464 GPU Hours |
| BiSSL Training | 83 GPU Hours |
| Fine-Tuning | 1-9 GPU Hours |

for $i = 1, \ldots, 10$. During the evaluation, it is important to note that the batch normalization layers (Ioffe & Szegedy, 2015) of the backbones utilize the running means and variances inferred during training. Since the pretext and BiSSL backbones $\boldsymbol{\theta}$, $\boldsymbol{\theta}_P^*(\boldsymbol{\theta}_D)$ have not been exposed to the downstream datasets during training, their batch normalization statistics may likely be suboptimal for the downstream datasets. To address this, the downstream training dataset is divided into batches of 256 samples, and roughly 100 batches are then forward-passed through these backbones, prior to calculating the output features in (13). This procedure updates the BN statistics to better reflect the downstream data, enabling a fairer comparison of the learned representations. The resulting feature-wise cosine similarities for each downstream dataset are reported in the right column of Table 9.

### B.10.2 Visual inspection of Latent Features

Test data features of the downstream test data processed by backbones trained through conventional pretext pretraining with SimCLR are compared against those additionally trained with BiSSL. This allows for an inspection of the learned representations prior to the final fine-tuning stage.

In same manner as in Section B.10.1, we first calibrate the batch normalization running statistics by forward-passing roughly 100 downstream training batches through the backbones. We then extract the backbone output features for all samples in the test partition, and reduce their dimensionality to two using t-Distributed Stochastic Neighbor Embedding (t-SNE) (Cieslak et al., 2020). Additional plots on other downstream datasets are provided in Section C.7.

## C  Additional Results

### C.1  Computation Times

We report the training times required for both pretext pretraining and BiSSL under the experimental setup described in Section 4.2.1 and 4.2.2. SimCLR pretraining was conducted on 8x A100 GPUs, and BiSSL training was measured on the DTD dataset using 4x A40 GPUs. Since the number of gradient steps is kept constant across all datasets in the main experiments, this measurement is representative of BiSSL's computational cost across other datasets as well. For transparency, we also report fine-tuning training times. The duration naturally varies with dataset size, and we present the range of computation times across the datasets listed in Table 1, measured on a single A40 GPU.

The reported training times reflects the total training time required on a single GPU, therefore we adjust for multi-GPU usage (e.g., for BiSSL, we multiply the total time taken by 4, as we used 4 GPUs). The results are documented in Table 10. BiSSL incurs some additional overhead over fine-tuning, but requires only a small fraction of the computational cost of pretraining.

### C.1.1  Remarks on Hessian Vector Products

A significant portion of the computational overhead in BiSSL arises from calculating Hessian-vector products (HVPs) during each upper-level gradient update. For each upper-level gradient update step, the upper-level

Table 11: Comparison of top-5 classification accuracies between the conventional SSL pipeline and our proposed BiSSL pipeline. Accuracies that are significantly higher from their counterparts are marked in bold font. Table 1 outlines the top-1 accuracies (and 11-point mAP for the VOC07 dataset).

| | Food | CIFAR10 | CIFAR100 | CUB200 | SUN397 | Cars | Aircrafts | DTD | Pets | Caltech101 | Flowers |
|---|---|---|---|---|---|---|---|---|---|---|---|
| **SimCLR:** | | | | | | | | | | | |
| FT | $91.3 \pm 0.1$ | $100.0 \pm 0.0$ | $96.4 \pm 0.1$ | $75.2 \pm 0.4$ | $79.5 \pm 0.4$ | $93.7 \pm 0.2$ | $81.9 \pm 0.5$ | $85.8 \pm 0.6$ | $94.2 \pm 0.4$ | $98.1 \pm 0.1$ | $94.5 \pm 0.2$ |
| BiSSL+FT | $\mathbf{93.4 \pm 0.0}$ | $100.0 \pm 0.0$ | $\mathbf{96.6 \pm 0.2}$ | $\mathbf{84.2 \pm 0.2}$ | $\mathbf{81.2 \pm 0.2}$ | $93.7 \pm 0.2$ | $\mathbf{84.3 \pm 0.3}$ | $\mathbf{87.8 \pm 0.3}$ | $\mathbf{96.1 \pm 0.1}$ | $\mathbf{98.6 \pm 0.1}$ | $\mathbf{95.3 \pm 0.2}$ |
| Avg Diff | $\mathbf{+2.1}$ | $0.0$ | $\mathbf{+0.2}$ | $\mathbf{+9.0}$ | $\mathbf{+1.7}$ | $0.0$ | $\mathbf{+2.4}$ | $\mathbf{+2.0}$ | $\mathbf{+1.9}$ | $\mathbf{+0.5}$ | $\mathbf{+0.8}$ |
| **BYOL:** | | | | | | | | | | | |
| FT | $91.9 \pm 0.1$ | $100.0 \pm 0.0$ | $96.3 \pm 0.1$ | $77.5 \pm 0.4$ | $77.8 \pm 0.3$ | $93.9 \pm 0.2$ | $81.8 \pm 0.6$ | $84.9 \pm 0.5$ | $95.6 \pm 0.2$ | $98.0 \pm 0.1$ | $93.6 \pm 0.3$ |
| BiSSL+FT | $\mathbf{92.3 \pm 0.1}$ | $100.0 \pm 0.0$ | $96.4 \pm 0.1$ | $\mathbf{84.5 \pm 0.2}$ | $\mathbf{79.7 \pm 0.1}$ | $94.0 \pm 0.2$ | $\mathbf{86.2 \pm 0.3}$ | $86.7 \pm 0.4$ | $96.6 \pm 0.2$ | $98.5 \pm 0.1$ | $94.2 \pm 0.1$ |
| Avg Diff | $\mathbf{+1.3}$ | $0.0$ | $+0.1$ | $\mathbf{+7.0}$ | $\mathbf{+1.9}$ | $+0.1$ | $\mathbf{+4.4}$ | $\mathbf{+1.8}$ | $\mathbf{+1.0}$ | $\mathbf{+0.5}$ | $\mathbf{+0.6}$ |

compute $N_c$ HVPs (five in the experiments of this paper) using the CG algorithm. Each HVP requires an additional backward pass, effectively adding $2N_c$ backward passes. Memory usage is also increased, as activations must be stored for the second backward pass.

The memory and runtime costs depend on the HVP implementation, typically requiring two to three times the memory of standard gradient computations and taking between two and four times longer to compute (Dagréou et al., 2024). Further multiplying the runtime values by $N_c$ highlights the substantial additional compute required by upper-level optimization relative to standard gradient computation. In principle, memory and runtime are expected to scale roughly linearly with backbone size. Under heavy memory load, however, additional runtime overhead can occur, potentially resulting in super-linear scaling for very large models. The latter could however be mitigated by combining BiSSL with parameter-efficient fine-tuning (PEFT) techniques, such as LoRA Hu et al. (2022), on both the upper and lower-levels, which we leave for future work.

## C.2 Top-5 Classification Accuracies

Table 11 outlines the corresponding top-5 accuracies of the experiments described in Section 4.3, which re-emphasizes the performance improvements imposed by BiSSL as initially implied by Table 1.

## C.3 Impact of Varying Upper-Level Iterations

BiSSL deviates from conventional bilevel optimization implementations by allowing multiple upper-level gradient updates before alternating back to the lower level, controlled via the hyperparameter $N_U$ in Algorithm 1. We claim that conducting additional upper-level iterations improves convergence efficiency during training.

Experiments were conducted using the same five downstream datasets used in Section 4.4.1 namely Pets, DTD, VOC07, Flowers and CUB200. As discussed there, these datasets span both coarse- and fine-grained classification tasks across diverse and largely disjoint domains. Also consistent with that setup, we use SimCLR as the pretext task.

We conduct experiments with $N_U = 1$ to match conventional bilevel update schedules, while keeping all other aspects identical to the default BiSSL setup in Section 4.2.2, aside from performing the subsequent fine-tuning hyperparameter grid search on 50 different hyperparameter combinations. Since $N_U = 1$ results in fewer total upper-level updates, we also experiment with a longer training duration by increasing the number of training stage alternations from $T = 500$ to $T = 4000$, matching the total number of gradient updates conducted in the default setup. Notice however that this results in significantly more number of total lower-level steps ($20 \cdot 4000 = 80,000$) conducted compared to the default setting ($40 \cdot 500 = 20,000$). Despite this, the results in Table 12 show that neither of the $N_U = 1$ variants provide any clear general advantage over the default configuration in the bottom row with $N_U = 8$ and $T = 500$.

Table 12: Impact on downstream classification accuracy from setting $N_U = 1$. All experiments use SimCLR as the pretext task. The bottom row ($N_U = 8, T = 500$) corresponds to the default BiSSL configuration described in Section 4.2.2.

| **BiSSL+FT** | Pets | DTD | VOC07 | Flowers | CUB200 |
|---|---|---|---|---|---|
| $N_U = 1, T = 500$ | $76.1 \pm 0.5$ | $61.8 \pm 0.4$ | $71.0 \pm 0.1$ | $82.4 \pm 0.5$ | $50.0 \pm 0.9$ |
| $N_U = 1, T = 4000$ | $78.0 \pm 0.2$ | $63.2 \pm 0.3$ | $71.4 \pm 0.1$ | $84.4 \pm 0.2$ | $56.0 \pm 0.4$ |
| $N_U = 8, T = 500$ | $78.3 \pm 0.2$ | $64.2 \pm 0.4$ | $71.5 \pm 0.1$ | $84.1 \pm 0.2$ | $55.7 \pm 0.2$ |

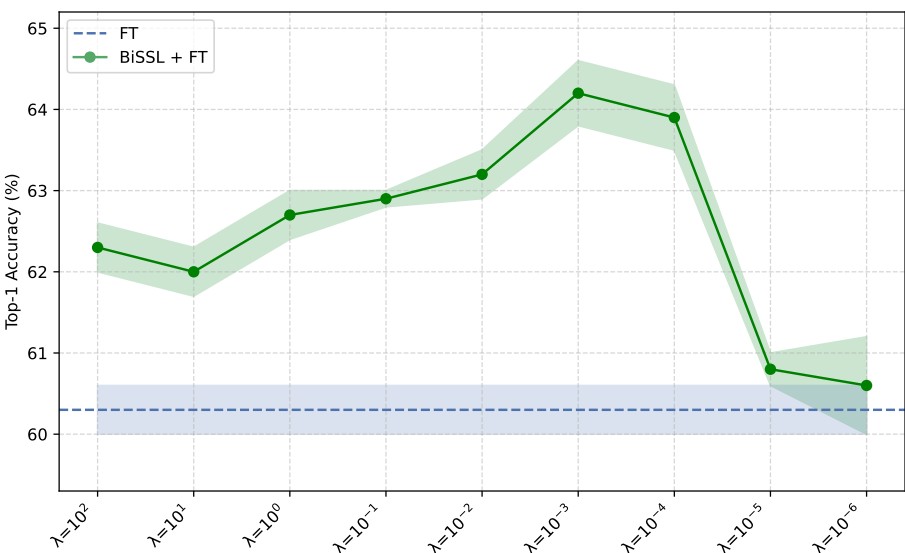

Figure 5: Top-1 test accuracies on the DTD dataset for models pretrained with SimCLR across different values of $\lambda$ used in the BiSSL training stage. When $\lambda$ is sufficiently large, BiSSL consistently yields substantial performance gains, with additional improvements possible through tuning around $\lambda = 0.001$.

### C.4  Impact of Varying the Lower-Level Regularization Weight

As discussed in Section 3.3, the magnitude of the lower-level regularization weight $\lambda$ controls the extent to which the upper-level downstream objective influences the lower-level pretext optimization. All main experiments in this paper use a constant value of $\lambda = 0.001$. To examine the effect of this parameter more systematically, we repeat the default experimental setup described in Section 4.2 while varying $\lambda$. For each value, we run the full fine-tuning hyperparameter search spanning 24 configurations. We use SimCLR as the pretext task and report results on the DTD dataset in Figure 5.

The results show that BiSSL provides notable improvements whenever $\lambda$ is large enough for the upper-level signal to meaningfully shape the lower-level optimization. More precise tuning yields further gains, with $\lambda = 0.001$ achieving the highest accuracy among the tested values. This outcome aligns with the interpretation in Section 3.3: When $\lambda$ is too small, the lower-level effectively behaves like continued pretext pretraining, providing little to no enhanced alignment with the downstream task. Furthermore, the theoretical setup in Section A.1 requires $\lambda$ to be sufficiently large in order to preserve lower-level convexity. Values of $\lambda$ that are too small can violate these assumptions, which may also help explain the negligible or absent performance gains observed in that regime.

Table 13: Top-1 validation classification accuracy on ImageNet using 1%, 10%, and 100% of the labeled training data. SimCLR is used as the pretext task. Accuracies statistically significantly higher from their counterparts are marked in the bold.

|  | ImageNet 1% | ImageNet 10% | ImageNet 100% |
|---|---|---|---|
| FT | $29.7 \pm 0.1$ | $50.9 \pm 0.1$ | $68.2 \pm 0.1$ |
| BiSSL+FT | $\mathbf{31.5 \pm 0.1}$ | $\mathbf{51.1 \pm 0.1}$ | $68.3 \pm 0.1$ |
| *Avg Diff* | $+\mathbf{1.8}$ | $+\mathbf{0.2}$ | $+0.1$ |

Table 14: Downstream classification performance using 224px images with the official pretrained backbone via SimCLR. BiSSL improves accuracy with statistical significance across four out of the five datasets.

| **SimCLR 224px** | Pets | DTD | VOC07 | Flowers | CUB200 |
|---|---|---|---|---|---|
| FT | $87.0 \pm 0.3$ | $69.5 \pm 0.4$ | $75.1 \pm 0.1$ | $89.7 \pm 0.3$ | $66.0 \pm 0.2$ |
| BiSSL+FT | $\mathbf{87.7 \pm 0.2}$ | $\mathbf{70.7 \pm 0.3}$ | $\mathbf{77.2 \pm 0.1}$ | $89.8 \pm 0.3$ | $\mathbf{71.6 \pm 0.3}$ |
| *Avg Diff* | $+\mathbf{0.7}$ | $+\mathbf{1.2}$ | $+\mathbf{2.1}$ | $+0.1$ | $+\mathbf{5.6}$ |

## C.5 Fine-Tuning on ImageNet

We expect the gains from BiSSL on ImageNet classification (Deng et al., 2009) as a downstream task to be limited in this specific setting, as pretraining is also conducted on ImageNet, which minimizes the distribution discrepancy between the pretext and downstream tasks. Nonetheless, for the sake of transparency, we conducted fine-tuning experiments on ImageNet within the bounds of our available computational resources.

Our setup consists of 50 fine-tuning epochs, using a hyperparameter grid search over 25 configurations. For BiSSL, we use $T = 100$ training stage alternations, while keeping the rest of the experimental setup consistent with the description in Section 4.2.2. We use SimCLR as the pretext task. Following standard practice in SSL literature for ImageNet fine-tuning (Chen et al., 2020; Grill et al., 2020; Bardes et al., 2022), we report the top-1 validation accuracy on models trained with the 1% and 10% subsets using the official splits from Chen et al. (2020). Additionally, we also include results on the full ImageNet dataset (100%).

Table 13 shows the results of these experiments. The most notable gain in the 1% setting may be attributed to a greater distribution mismatch between the small labeled subset and the pretraining dataset, representing the type of scenario where BiSSL is particularly effective at improving alignment. In contrast, the 10% subset is more representative of the full dataset, and the relative advantage of BiSSL correspondingly decrease.

## C.6 Larger Image Size with Official SimCLR Backbone Weights

To assess how BiSSL scales to higher image resolutions and to verify its compatibility with off-the-shelf pretrained models, we benchmark BiSSL using the official SimCLR ResNet-50 backbone weights released by Chen et al. (2020), trained with an input size of $224 \times 244$.

Because the official SimCLR checkpoints do not include their original projection head, we trained a replacement projection head with a similar architecture on the frozen backbone for 20 epochs using the SimCLR objective. To accommodate computational limits, we used a batch size of 1024 instead of the original 4096. For BiSSL, we further reduced total training time by setting $T = 200$, $N_U = 4$, and $N_L = 10$ (see Algorithm 1 for details). The subsequent fine-tuning hyperparameter grid search is conducted over 25 different combina-

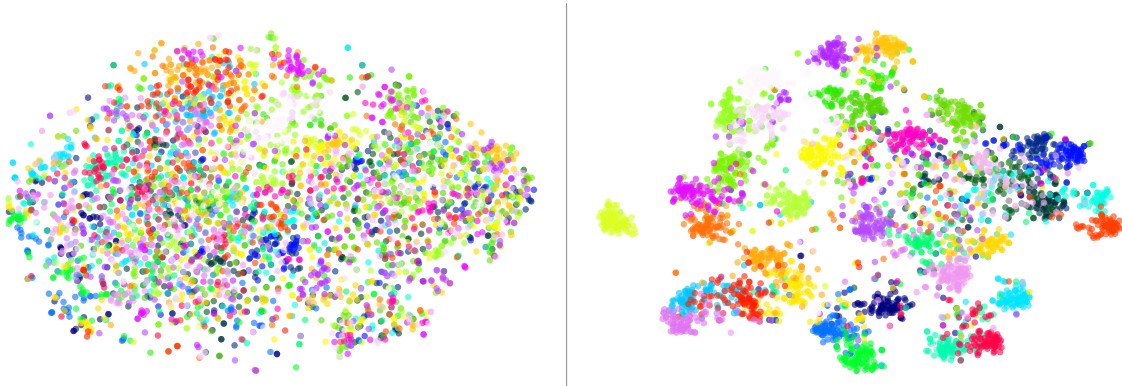

Figure 6: Features from backbones after applying self-supervised pretraining (left) and BiSSL (right) on the Pets (Parkhi et al., 2012) dataset.

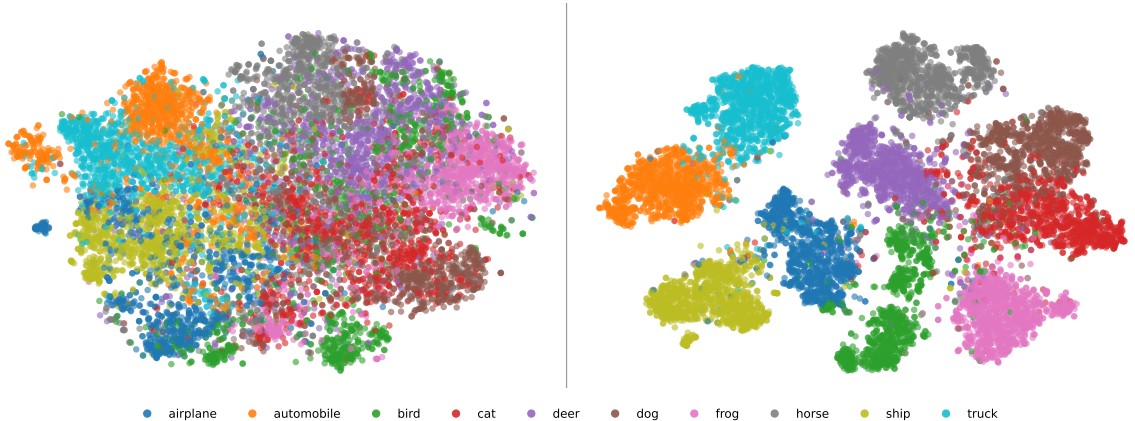

Figure 7: Features from backbones after applying self-supervised pretraining (left) and BiSSL (right) on the CIFAR10 (Krizhevsky, 2012) dataset.

tions. We used the larger input resolution of $244 \times 244$, with all other settings identical to those described in Sections 4.2.1 and 4.2.2. Experiments were performed on the same five downstream classification dataset used in Section 4.4.1, namely Pets, DTD, VOC07, Flowers and CUB200. As argued in that section, these datasets cover both coarse- and fine-grained classification tasks across diverse and mostly disjoint domains.

While the setup does not replicate the full-scale SimCLR configuration (lacking the original projection, batch size, and employing a shorter BiSSL training duration), it nonetheless provides a meaningful indicator of BiSSL's robustness. As shown in Table 14, BiSSL sustains its ability to significantly enhance downstream performance, even under these conditions.

### C.7 Visual inspection of Latent Features

Figure 4 illustrated features on the flowers dataset from SimCLR-pretrained backbones to BiSSL-trained backbones. Figures 6 to 11 illustrate similar backbone features on another selection of the downstream datasets described in Section 4.1. The trend, where BiSSL appear to better align the backbone with the downstream task, persist even for datasets where the accuracy gains from applying BiSSL are minimal or absent.

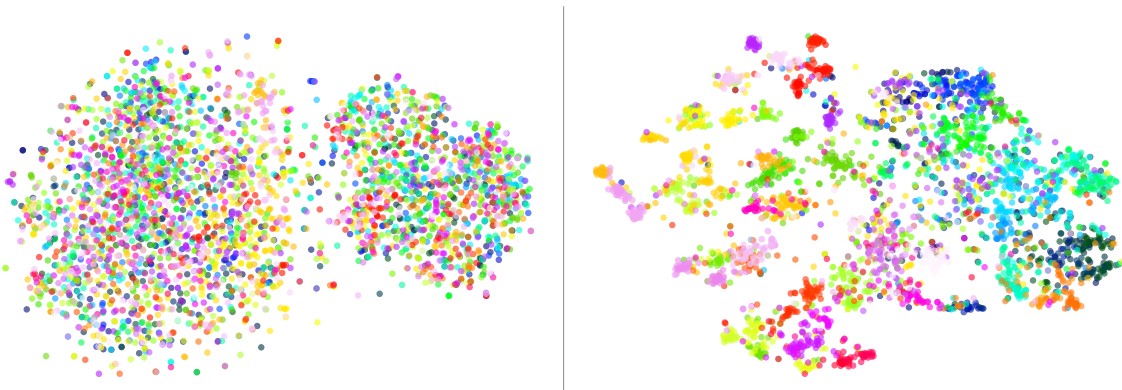

Figure 8: Features from backbones after applying self-supervised pretraining (left) and BiSSL (right) on the Aircrafts (Maji et al., 2013) dataset.

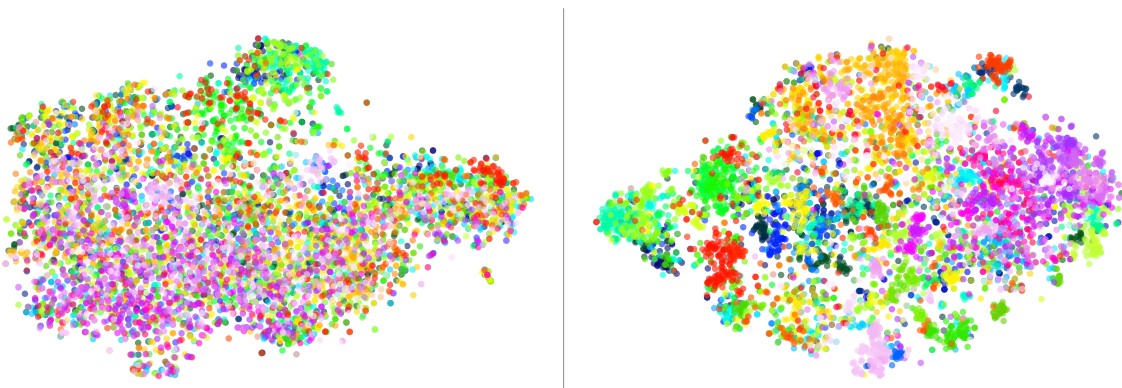

Figure 9: Features from backbones after applying self-supervised pretraining (left) and BiSSL (right) on the CUB200 (Wah et al., 2011) dataset.

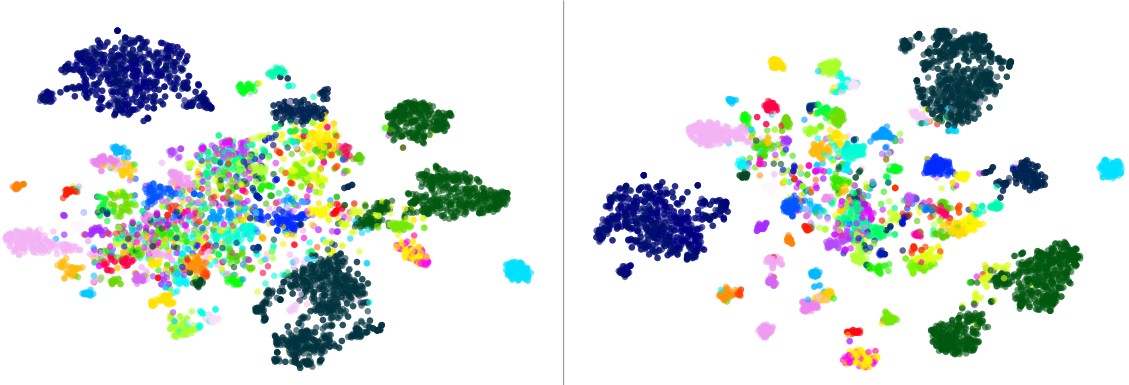

Figure 10: Features from backbones after applying self-supervised pretraining (left) and BiSSL (right) on the Caltech-101 (Li et al., 2022a) dataset.

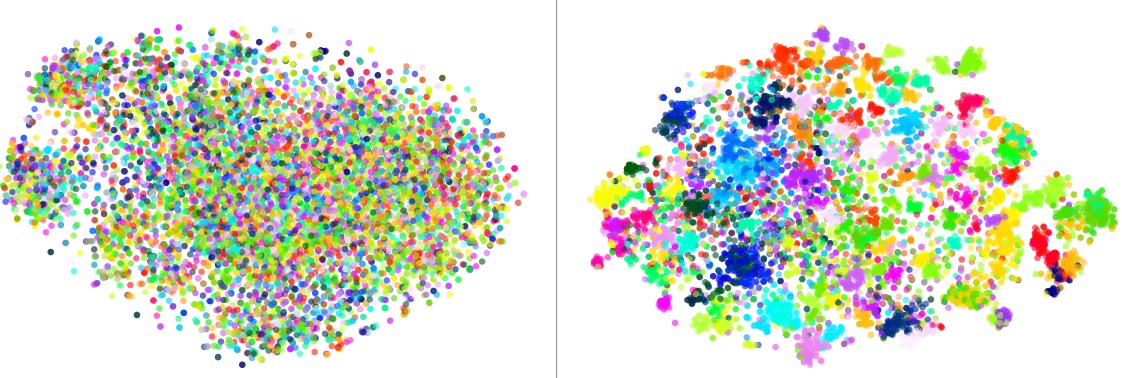

Figure 11: Features from backbones after applying self-supervised pretraining (left) and BiSSL (right) on the Cars (Yang et al., 2015) dataset.

## D   Discussion of Limitations and Future Work

The bilevel optimization procedure of this study employs a conjugate gradient (CG) solver, selected for its relative simplicity and prior success in related settings (Rajeswaran et al., 2019; Pedregosa, 2016). While this solver proved effective in our experiments, it may not be the most optimal in terms of computational efficiency or compatibility with the non-convex landscape inherent in deep neural networks. Future work could explore alternative solvers (Huang, 2024; Hao et al., 2024; Yang et al., 2025), approximation methods for the inverse Hessian vector products (Singh & Alistarh, 2020; Frantar et al., 2021) or reformulations of the optimization problem to further improve both accuracy and efficiency.

Access to the pretext task, pretraining data, and associated pretext head parameters are current requirements for applying BiSSL. This may limit applicability in some constrained environments, although future work could investigate strategies to partially or completely decouple BiSSL from these dependencies.

BiSSL also requires the backbone architectures used in the pretext and downstream stages to be identical. While this is standard in many transfer learning pipelines, it may limit flexibility in scenarios involving architectural modifications, such as the use of parameter-efficient fine-tuning methods (Hu et al., 2022). However, this constraint could be mitigated by applying the same architectural modifications consistently across both stages of the BiSSL framework, another avenue worth exploring in future work.

Finally, the benefits from BiSSL are expected to be most pronounced when there is a meaningful distributional discrepancy between the pretraining and downstream tasks. As demonstrated in Section C.5, when the downstream data distribution closely aligns with the pretext task, BiSSL may offer limited improvement. In practice, some degree of distributional shift is often present, but users should be mindful that the effectiveness of BiSSL may depend on the extent of that mismatch.

