# OpenReview forum: "BiSSL: Enhancing the Alignment Between Self-Supervised Pretraining and Downstream Fine-Tuning via Bilevel Optimization"
_TMLR — Accepted by TMLR_

### Review · Reviewer_5F7N · 2025-11-22

**Summary Of Contributions:**

The paper introduces BiSSL, a bilevel optimization framework intended to better align self-supervised pretrained backbones with downstream fine-tuning tasks. Instead of relying purely on pretext-only pretraining, the authors insert a preparatory training stage where the pretext objective (lower level) and the downstream objective (upper level) jointly influence the backbone parameters. The lower level is regularized to stay close to the upper-level backbone, and the upper level uses an implicit-gradient approximation via conjugate gradients to guide the pretext optimization toward representations that are ultimately more compatible with the downstream task.

Empirically, the paper evaluates BiSSL very extensively—on SimCLR, BYOL, MAE; on ResNet and ViT architectures; and across many downstream classification, detection, and segmentation datasets. The method consistently improves fine-tuning performance and reduces variance. Additional analyses measuring cosine similarity and t-SNE visualizations also support the claim that BiSSL produces backbones that are more stable under fine-tuning.

Strengths
- Very comprehensive experimental evaluation across many datasets and architectures.
- The bilevel formulation is conceptually clean and methodologically interesting.
- Consistent and statistically significant performance gains.
- The alignment analyses are simple but effective and help motivate the method.

Weaknesses
- The method adds a fairly heavy training stage (multiple alternations, CG steps), and computational overhead is not fully addressed.
- Several training choices (e.g., warm-up, λ selection, NL/NU schedule) feel somewhat ad-hoc and might require expertise to tune.
- The theoretical section mainly formalizes the optimization but does not provide much deeper insight into why BiSSL should work beyond empirical evidence.

**Audience:**

Yes

**Audience Explanation:**

TMLR readers working on self-supervised learning, meta-learning, bilevel optimization, and transfer learning will all find the idea relevant. The problem of pretraining–downstream misalignment is widely recognized, and a general-purpose framework that can sit between pretraining and fine-tuning is timely. The empirical improvements alone should be of interest to practitioners.

**Broader Impact Concerns:**

The paper does not appear to introduce new societal or ethical risks beyond standard self-supervised learning. It may reduce reliance on large labeled datasets, which is usually positive. There is no application domain discussed that would raise special concerns. A brief Broader Impact section acknowledging this would be sufficient.

**Claims And Evidence:**

Yes

**Claims Explanation:**

The claims about improved downstream performance are well supported: the paper shows results across 12+ classification datasets, two detection/segmentation tasks, three different pretext tasks, and two backbone families. The gains are generally consistent and statistically significant. The ablations and extended experiments lend credibility to the robustness of the approach.

The “better alignment” narrative is necessarily more qualitative, but the cosine-similarity results and feature visualizations provide reasonable evidence that the method makes the pretrained model more stable with respect to fine-tuning.

My only hesitation is that the method’s sensitivity to various hyperparameters is not extensively analyzed, but this does not substantially detract from the strength of the empirical results.

**Requested Changes:**

Critical

Discuss computational cost more explicitly. The extra bilevel stage and CG computations likely add nontrivial overhead. A clearer breakdown and comparison with baseline pretraining cost would help readers judge practicality.

Non-critical (would strengthen the work)

Clarify the intuition for the intermediate λ regime beyond the formal derivation.

Provide more detail on failure cases or datasets where BiSSL provides only marginal improvement.

Consider adding a lighter-weight variant of BiSSL or discussing how one might reduce CG overhead.

Improve the readability of Algorithm 1 with a higher-level explanation before diving into the full pseudocode.

---

> ### Author Response · Authors · 2025-12-11
> **Author's Response to Reviewer 5F7N**
>
> We thank the reviewer for taking the time to evaluate our paper. We are pleased with the positive assessment of our work, particularly the recognition of our method as both conceptually clear and interesting, as well as the extensiveness of our evaluation. Below, we address the specific concerns raised and outline the corresponding updates made to the submission.
>
> ---
>
> ## Critical Requested Changes
> ### Discussion of Computational Costs
> We thank the reviewer for raising this point regarding documentation of computational cost. We provide respective compute times for the pretext, BiSSL, and fine-tuning stages in Section C.1 of the appendix. Additionally, our response to Reviewer 6YHH provides a more detailed breakdown of the most computationally expensive overhead introduced from Hessian-vector product calculations. These insights have also been incorporated into Section C.1 of the submission.
>
> We hope that these results and discussions adequately address the reviewer’s concern, otherwise we are happy to provide further details.
>
> ---
>
> ## Non-Critical Requested Changes
> ### Clarification on Intuition for $\lambda$
> In the last paragraph of Section 3.3, we interpret how BiSSL optimization behaves at extreme values of $\lambda$, supporting the conclusion that an intermediate value balances contributions from both levels. Additionally, in response to Reviewer HMne, we now also include a sensitivity analysis on the DTD dataset, benchmarking performance under varying $\lambda$ (added to Section C.4 of the Appendix). We believe this captures the main conceptual insight, but remain happy to provide further discussion or clarification if helpful.
>
> ### Datasets With Only Marginal Improvements from BiSSL
> Thank you for this suggestion. As BiSSL is designed to better align pretrained models with downstream tasks, we noted in the discussion of limitations in Section D of the appendix that improvements are more marginal when the downstream task is already relatively well-aligned with the pretext task. To further elaborate, datasets such as CIFAR10+100, VOC07 and Caltech101 likely benefit less from BiSSL because they are coarse-grained tasks with input distributions that are more closely matching the ImageNet image distribution. By contrast, fine-grained datasets that require more specialized features and have more distinct input distributions, such as CUB200, Aircrafts, and Pets, show larger improvements from BiSSL. We have updated the results section of the main body to include this elaboration.
>
>
> ### Light-Weight Variant
> Developing a light-weight variant of BiSSL is indeed a relevant and valuable direction. It is part of our ongoing work, which has shown promising initial results, but including this in detail would go beyond the scope of this paper. For now, we have noted in Appendix D that the current approach may not be optimal in terms of computational efficiency, and suggest that future work could explore alternative solvers, approximation methods or reformulations to improve upon this.
>
>
> ### Readability of Algorithm 1
> We thank the reviewer for this suggestion. We have revised the description to first provide a high-level overview of the key aspects of the proposed algorithm before referring the reader to the actual pseudocode.
>
> ---
>
> ## Broader Impact Statement
> We have added a brief statement in the Broader Impact section acknowledging the concern highlighted.
>
> ---
>
> ## Conclusion
> We thank the reviewers again for their valuable feedback. We believe the updates and clarifications provided have strengthened the paper, and we remain happy to provide further details or discussion if needed.

---

> > ### Comment · Reviewer_5F7N · 2026-01-05
> >
> > Thank you. I am satisfied with the response and have no further comments.

---

### Review · Reviewer_HMne · 2025-12-03

**Summary Of Contributions:**

The paper proposes BiSSL, a bilevel optimization framework that couples self-supervised pretext objective and downstream supervised objective during an intermediate stage between pretraining and fine-tuning. The method uses implicit differentiation to let the downstream objective shape the pretext backbone via a similarity regularizer and the implicit Jacobian, and then initializes final fine-tuning from the lower-level solution. Across SimCLR or BYOL pretraining on ImageNet and multiple downstream tasks (12 classification datasets, VOC/Cityscapes detection/segmentation, and ViT+MAE), BiSSL yields consistent, significant improvements over standard fine-tuning and several strong baselines, with analyses indicating improved downstream alignment.

**Audience:**

Yes

**Audience Explanation:**

1. The work is at the intersection of self-supervised learning, transfer learning, and bilevel optimization, all of which are ongoing topics of interest for the TMLR audience.
2. The work is important, for many works use off-the-shelf SSL backbones and care about how to adapt them to downstream tasks without changing pretraining.

**Broader Impact Concerns:**

Ethical risks are not present, as BiSSL does not introduce new data sources or label types. Its is a training procedure applied on existing datasets commonly used in vision.

**Claims And Evidence:**

Yes

**Claims Explanation:**

Novelty:
1. The bilevel setup that connects the SSL pretext stage with downstream fine-tuning is clearly motivated. It is different from earlier BLO approaches that operate only at pretraining or only at fine-tuning.
2. The upper-level gradient derivation follows the general idea of iMAML-style implicit differentiation, but here it is adapted to the SSL plus fine-tuning pipeline, and the analysis helps explain how the similarity regularizer and its scaling parameter influence the interaction between the two stages.
3. The  training procedure is task-agnostic and works with different SSL methods, backbones (ResNet and ViT), and downstream heads.
Experiments:
1. The experiments cover a wide range of settings: 12 classification datasets, detection, and segmentation, using both SimCLR and BYOL. The improvements are consistent, and the variance is reduced.
 2. The comparisons include several baselines such as continued pretraining on mixed data, joint loss (pretext+downstream), NoisyTune, fine-tuning variants, and a BiSSL variant that removes the IJ coupling.
3. Additional results on ViT with MAE support the method’s generality. The authors also provide useful ablations (for example on the number of upper-level steps) and include statistical significance tests.

**Requested Changes:**

The below proposed adjustments are for strengthening the work:
1. Section 3.3, I would suggest to move the derivation to the appendix and keep the focus of the section on the high-level intuition.
2. State more clearly that the theoretical assumptions  might not strictly hold in deep networks.
3. Provide the sensitivity curve of regularization weight on atleast one or two datasets.

---

> ### Author Response · Authors · 2025-12-11
> **Author's Response to Reviewer HMne**
>
> Dear Reviewer,
>
> Thank you for your comments. We are grateful for the overall positive assessment of our work. We especially appreciate that our method is recognized as sound and novel, that its generality is acknowledged, and that the thoroughness of our evaluation is noted. In the response below, we address the suggested adjustments aimed at further strengthening the work.
>
> ---
>
> ## Suggested Changes
>
> ### RC1: Focus on High-Level Intuition in Section 3.3
> To improve readability, we added a line after the derivations in Section 3.3 providing a high-level explanation of how the pretext objective is explicitly incorporated into the upper-level gradient. Most of the derivations leading to the final gradient expression are already deferred to the Appendix in Section A.1, and we respectfully believe that the remaining ones in the main body help clarify more explicitly how the two levels are connected in BiSSL while also serving as an insight into how its gradients are calculated in practice.
>
>
> ### RC2: Emphasis on Validity of Theoretical Assumptions in Practice
> Thank you for pointing this out. We have added a note in Appendix A.1 clarifying that the theoretical assumptions may not strictly hold in deep networks, but that the resulting expressions have nonetheless proven practically beneficial in both our and prior work.
>
>
> ### RC3: Sensitivity Curve of Regularization Weight
> We agree that including an ablation examining the impact of varying $\lambda$ would make the work more complete. Accordingly, we have run such experiments during the rebuttal. The BiSSL setup remain identical to the core setup of the paper, with the only difference being the variation of $\lambda$. For each value of $\lambda$, we conduct the fine-tuning hyperparameter grid search over $24$ different combinations. We use SimCLR as the pretext task. Results on the DTD dataset are shown in the table below, where we marked the value $\lambda=0.001$ used throughout all other experiments of the paper with bold. The last row shows the baseline fine-tuning accuracy for ease of comparison.
>
> |                            |     Top-1 Accuracy      |
> | :------------------------- | :---------------------: |
> | $\lambda=10^{2}$           |     $62.3 \pm 0.3$      |
> | $\lambda=10^{1}$           |     $62.0 \pm 0.3$      |
> | $\lambda=10^{0}$           |     $62.7 \pm 0.3$      |
> | $\lambda=10^{-1}$          |     $62.9 \pm 0.1$      |
> | $\lambda=10^{-2}$          |     $63.2 \pm 0.3$      |
> | $\mathbf{\lambda=10^{-3}}$ | $\mathbf{64.2 \pm 0.4}$ |
> | $\lambda=10^{-4}$          |     $63.9 \pm 0.4$      |
> | $\lambda=10^{-5}$          |     $60.8 \pm 0.2$      |
> | $\lambda=10^{-6}$          |     $60.6 \pm 0.6$      |
> | -------------              |     -------------       |
> | FT                         |     $60.3 \pm 0.3$      |
>
>
> BiSSL maintains a substantial performance gap over the fine-tuning baseline when $\lambda$ is sufficiently large, with the best results at $\lambda=0.001$. This demonstrates that BiSSL provides significant gains even without cautiously selecting $\lambda$, though performance can be further improved with more careful tuning. For the very small values of $\lambda$, we hypothesize that the upper-level exerts too little influence on the lower-level, effectively reducing it to an extended self-supervised pretraining stage, which accounts for the near baseline accuracies observed. We have added all experimental details to Section C.4 of the Appendix, where the results are presented as a curve for a clearer presentation.
>
> ---
>
> ## Conclusion
> We again thank the reviewer for their feedback, and believe the suggested changes have helped further strengthen the paper. We remain happy to provide further details or discussion if requested.

---

> > ### Comment · Reviewer_HMne · 2026-01-05
> >
> > I am satisfied with your response and have no further questions.

---

### Review · Reviewer_6YHH · 2025-12-06

**Summary Of Contributions:**

This paper introduces a framework for aligning self-supervised pretraining and downstream tasks through a bilevel optimization (BLO) formulation. The main motivation for using BLO is that the pretext task is explicitly made downstream-task-dependent: the downstream objective is optimized at the upper level, while the pretext task is optimized at the lower level within the BLO framework.

Although BLO has previously been used in self-supervised learning, the formulation in this paper is pretext-task agnostic, i.e., it can be combined with existing SSL methods such as SimCLR. This modularity is a strong point of the work and could make the approach broadly applicable and impactful within the SSL community.

The formulation of the BLO problem and the proposed algorithm is clearly presented. The paper provides explicit derivations of the upper- and lower-level gradients required for the algorithmic design. Building upon prior work on BLO, the paper also states assumptions on the convexity of the lower-level optimization problem in order to guarantee stationarity conditions.

Although the method demonstrates lower runtime than the baseline SSL methods in the reported experiments, a notable weakness is its scalability to larger backbone models. As mentioned in the appendix, the experiments rely on lightweight ViT architectures (for which the parameter count is not reported), and the implicit Jacobian computation and storage scale approximately linearly with the number of parameters. This raises concerns about the practicality of the approach for large-scale models commonly used in modern SSL.

**Audience:**

Yes

**Audience Explanation:**

Yes. The paper should be of broad interest, as SSL is a central topic in the current machine learning community. In addition, the work presents interesting mathematical derivations of the BLO formulation, and the resulting algorithmic framework may also appeal to researchers in optimization and bilevel learning.

**Broader Impact Concerns:**

My main concern regarding the broader impact of the paper is the scalability of the method as the backbone model size increases. The paper should explicitly discuss this limitation and its implications for deploying the method in realistic large-scale settings. Additionally, the authors should elaborate on whether and how the learned pretext task generalizes to downstream tasks that differ from those used in the BLO stage, as this has direct consequences for the practicality and impact of the proposed framework in typical SSL use cases.

**Claims And Evidence:**

Yes

**Claims Explanation:**

The method is comprehensively evaluated across several datasets (Pets, DTD, VOC07, Flowers, CUB200, CIFAR10, CIFAR100, among others), compared with different SSL frameworks such as SimCLR, MAE, and BYOL, and across different tasks. The results show consistent performance improvements in all reported scenarios. The paper also clearly documents the hyperparameter settings for each experiment. Finally, an interpretable feature analysis of the pretext task is presented via 2D t-SNE visualizations, showing that the proposed pretext process produces feature representations that cluster image classes more distinctly than those obtained from baseline self-supervised pretraining.

However, one important issue regarding the evaluation is that it is unclear whether the BLO framework requires the downstream task used in the upper-level objective to be the same as the downstream task used during fine-tuning. This is critical because SSL methods are typically designed to be general-purpose and transferable across downstream tasks, whereas the proposed approach appears to be task-specific, potentially requiring separate pretraining stages for each distinct downstream task considered at fine-tuning.

**Requested Changes:**

The paper mentions that the lower-level optimization problem is convex since $r(\dot)$ is convex, but $L^P$ is clearly non-convex. The paper should clearly state under which conditions of $L^P$ the problem is convex (e.g., weak convexity, PL condition, or other structural properties) under which the overall lower-level problem becomes convex, or otherwise satisfies the conditions required for the theoretical guarantees and hypergradient derivations.


The paper should more explicitly characterize how memory consumption and computational time scale as the backbone size increases. This could include, for example, empirical measurements or complexity estimates that quantify the growth in runtime and memory when moving from lightweight ViTs to larger modern architectures

The paper should clearly state whether the downstream task used in the upper-level objective during the BLO stage is the same as the downstream task used for fine-tuning. If this is the case, the authors should discuss how the approach could be generalized to avoid requiring a distinct BLO-based SSL stage for every downstream task, and to what extent the learned pretext task transfers across different downstream tasks.

---

> ### Author Response · Authors · 2025-12-11
> **Author's Response to Reviewer 6YHH [Part 1/2]**
>
> We thank the reviewer for their assessment of our work. We are glad that the modularity of our method is recognized and found to be clearly presented, and we appreciate the acknowledgment of the extensive experimental evaluation. We also value the reviewer's view that the paper should be of broad interest in the machine learning community. Below, we address the concerns raised to the best of our ability.
>
> ### Remark on Backbone Architectures Utilized in the Experiments
> We would like to clarify a point mentioned in the summary, which states that the experiments rely on a ViT architecture. The ViT backbone is used only for the experiments in Section 4.4.2, while the majority of the remaining experiments utilize a ResNet-50. Regarding parameter counts, we follow the standard convention and specify in the experiments specification of Section B.8 that a ViT-S backbone is used, with a reference to the source detailing the specific hyperparameters. We note that this clarification does not diminish the reviewer’s concerns regarding scalability, which we address below, but we wanted to ensure the record is accurate.
>
> ## Requested Changes
> ### RC1: Conditions for Lower-Level Convexity
> Thank you for pointing this out, we agree that specifying more explicitly under which conditions the lower-level is convex should be more clearly stated. Denote the lower-level objective as $\mathcal{L}^P(\boldsymbol{\theta}_P) + \lambda r(\boldsymbol{\theta}_D,\boldsymbol{\theta}_P)$. Assuming the regularization objective $r$ is strongly convex, then if the pretext objective $\mathcal{L}^P$ is twice-differentiable and $L$-smooth, then there exists sufficiently large values of $\lambda$ which makes the lower-level objective convex. We have updated the theoretical deviations and assumptions of appendix A.1 to now reflect this.
>
>
> ### RC3: Downstream Task in Upper-Level Objective / Clarification of BiSSL Method
> The downstream objective $\mathcal{L}^D$ used in the upper-level BLO problem is the same as the one used in fine-tuning. We have added a remark to the training pipeline description in Section 3.4 to highlight this point.
>
>
> In this context, we would like to take the opportunity to clarify the role of BiSSL. It is not a method that replaces the SSL pretraining stage, nor does it require separate pretraining for each downstream task. BiSSL operates on a backbone that has already been pretrained with some standard task agnostic pretext objective, and this same pretrained backbone is reused across all downstream tasks. BiSSL then applies a significantly shorter training stage targeted to align the pretrained backbone with the downstream task before fine-tuning. We politely argue that there is hence no need to generalize BiSSL across multiple downstream tasks, as it is not a separate SSL stage but a preparatory adaptation stage to downstream fine-tuning of an already pretrained backbone.
>
> *Continued...*

---

> > ### Author Response · Authors · 2025-12-11
> > **Author's Response to Reviewer 6YHH [Part 2/2]**
> >
> > ### RC2: Memory and Time Complexity
> > We thank the reviewer for raising this valid point regarding the need for more explicit characterization of memory and time consumption. The main computational overhead in BiSSL arises from calculating Hessian-vector products (HVPs) during each upper-level gradient update. Specifically, for each upper-level gradient update step, we compute $N_c$ HVPs (five in our experiments) through the CG algorithm. Each HVP requires an additional backward pass, which effectively adds $2N_c$ backward passes. Memory usage is also higher because activations must be stored for the second backward pass.
> >
> > The actual memory and runtime costs depend on the HVP implementation as discussed in [this ICLR blogpost](https://iclr-blogposts.github.io/2024/blog/bench-hvp/), stating that HVPs require roughly two to three times the memory of standard gradients and take between two and four times longer to compute. Multiplying the runtime values by $N_c$ to reflect the additional cost of the upper-level gradient computation makes it clear that the upper-level optimization incurs a substantial runtime increase compared to standard gradient calculation. In principle, memory and runtime should grow roughly linearly with backbone size, but in practice additional runtime overhead can occur when memory is heavily utilized, potentially resulting in super-linear scaling for very large models. However, parameter efficient fine tuning methods such as LoRA are typically used with larger backbones to reduce memory overhead, and they can be applied on both the upper and lower-levels of BiSSL to mitigate this issue in the same way. We have added all insights in addition to the computation times listed in Section C.1 of the submission. *We also reemphasize that BiSSL is applied as a much shorter training stage on already pretrained backbones, serving merely as a downstream alignment step prior to fine-tuning, which keeps the additional runtime more manageable.*
> >
> > A light-weight variant of BiSSL that avoids full HVP computation would further mitigate this scalability issue. Developing such a variant is part of our ongoing work and has shown promising preliminary results, but lies beyond the scope of this paper.
> >
> >
> > ## Conclusion
> > We again thank the reviewer for their assessment, and find their feedback fruitful. We hope our response have addressed the concerns raised, and we stay available to address any further potential inquiries.

---

### Decision · Action_Editor_i12U · 2026-01-21

**Recommendation:** Accept as is

**Audience:**

Yes

**Audience Explanation:**

The methods proposed can be of interest for the related research community since the techniques introduced can better align two important stages in modern learning methods

**Claims And Evidence:**

Yes

**Claims Explanation:**

As described by the reviewers, the paper introduces novel methods for self-supervised pre-training and fine-tuning that can be of interest for the researchers in the field. In addition, the authors have adequately addressed the questions and comments raised by the reviewers in their reviews. The results are backed by some theoretical results and multiple strong experimental results